# Mineral Materials Coated with and Consisting of MnO_x_—Characteristics and Application of Filter Media for Groundwater Treatment: A Review

**DOI:** 10.3390/ma13102232

**Published:** 2020-05-13

**Authors:** Magdalena M. Michel, Lidia Reczek, Dorota Papciak, Maria Włodarczyk-Makuła, Tadeusz Siwiec, Yuliia Trach

**Affiliations:** 1Institute of Environmental Engineering, Warsaw University of Life Sciences—SGGW, 166 Nowoursynowska st., 02-787 Warsaw, Poland; lidia_reczek@sggw.edu.pl (L.R.); tadeusz_siwiec@sggw.edu.pl (T.S.); 2Faculty of Civil and Environmental Engineering and Architecture, Rzeszow University of Technology, 6 Powstańców Warszawy Ave., 35-959 Rzeszów, Poland; dpapciak@prz.edu.pl; 3Faculty of Infrastructure and Environment, Czestochowa University of Technology, 69 Dąbrowskiego st., 42-200 Częstochowa, Poland; mwm@is.pcz.czest.pl; 4Department of Water Supply, Water Disposal and Drilling Engineering, National University of Water and Environmental Engineering, 11 Soborna st., 33028 Rivne, Ukraine; trach1805@gmail.com

**Keywords:** drinking water, water treatment, surface coatings, quartz sand, chalcedonite, diatomite, glauconite, zeolite, pyrolusite, anthracite

## Abstract

For groundwater treatment, the technologies involving oxidation on MnO_x_ filter bed are beneficial, common, and effectively used. The presence of MnO_x_ is the mutual feature of filter media, both MnO_x_-coated mineral materials like quartz sand and gravel, chalcedonite, diatomite, glauconite, zeolite, or anthracite along with consisting of MnO_x_ manganese ores. This review is based on the analysis of research and review papers, commercial data sheets, and standards. The paper aimed to provide new suggestions and useful information for further investigation of MnO_x_ filter media for groundwater treatment. The presented compilations are based on the characteristics of coatings, methods, and conditions of its obtaining and type of filter media. The relationship between the properties of MnO_x_ amendments and the obtained purification effects as well as the commonly used commercial products, their features, and applications have been discussed. The paper concludes by mentioning about improving catalytic/adsorption properties of non-reactive siliceous media opposed to ion-exchange minerals and about possible significance of birnessite type manganese oxide for water treatment. Research needs related to the assessment of the use MnO_x_ filter media to heavy metals removal from groundwater in field operations and to standardize methodology of testing MnO_x_ filter media for water treatment were identified.

## 1. Introduction

Rocks and minerals are commonly used in various applications of environmental engineering. In water technology and protection, these are used as adsorbents and ion exchangers [1,2,3,4,5], filter media [6,7], remineralizing and deacidifying materials [8,9], coagulant aids [7], as well as substrates for the production of inorganic membranes [10,11,12] or sorbents for removal of petroleum substances [13]. In groundwater treatment, the filtration systems are a technical representation of the natural processes that occur between water components and rock material. It is primarily filtration through the porous material, when the suspended particles are removed as a result of screening, sedimentation, flocculation, adhesion, attraction, and repulsion. Depending on the type and properties of the mineral media, filtration may be accompanied by ion exchange, adsorption, chemisorption, and biosorption to remove water particles and ions. Quartz sand and gravel, most commonly used for the treatment of groundwater, are also the oldest filter materials. As technological development increased, the variety of minerals mined extended, and research methods were developed to characterize their properties, which resulted in the application of new materials. The effect of this is the use of filters with anthracite, chalcedonite, diatomite, glauconite, pyrolusite and zeolites on groundwater treatment plants. The specific physical and chemical properties of rocks and minerals determine their purpose in technological systems of groundwater treatment.

According to the World Health Organization guidelines for drinking water quality [14], its components cannot constitute a threat to human health. Groundwater can be a valuable resource of drinking water due to less pollution compared to surface water, though depending on the geological structure, hydrogeological and climatic conditions, in some regions of the world with intensive agricultural, urban and industrial land use [15,16,17] it is already contaminated with heavy metals and metalloids, nitrate, organics or halides [18,19,20,21,22,23,24,25,26,27]. Iron, manganese, ammonium ion and hydrogen sulphide are common components of groundwater and can be removed by classic technologies based on manganese oxides coated and consisting of filter media. The excessive concentration of these components leads to problems such as undesirable taste, odor, and color of water, staining laundry and clogging of fittings by sediment and biofilm formation [6]. Inorganic iron is easy to remove, whereas manganese, ammonium nitrogen, and hydrogen sulphide removal are definitely more arduous. Their removal also involves oxidation to hardly soluble iron and manganese oxyhydroxides retained in the filter bed, as well as to more acceptable nitrates. Reduced forms of iron and manganese can be removed by homogeneous oxidation (with air oxygen or oxidants such as chlorine or potassium permanganate) and hydrolysis leading to precipitation of its oxyhydroxides and subsequent deposition on and between filter medium grains [28,29]. In the presence of coated filter media, the oxidation of iron and manganese is a heterogeneous type consisting of the adsorption of metals on the surface of the coating, subsequent oxidation and retention on the surface [28,29,30]. Another mechanism for removing iron, manganese and mainly transforming ammonium ions is the biological oxidation by bacteria growing on the surface of the filter media [31]. Mostly, the removal of hydrogen sulfide from water involves desorption during aeration, but it can also be removed by oxidation to sulfur in the presence of a catalytic coating on a filter medium [32]. Methods based on oxidation on the surface of the filter bed in the presence of MnO_x_ are recommended as more beneficial solutions since they reduce the addition of chemicals into water, especially chlorine compounds, which prevents the formation of oxidation by-products [33]. The particular utility is due to the fact that under optimized conditions this kind of treatment does not lead to the formation of toxic residues which may arise in the use of chemical oxidants such as chlorine or ozone. Technologies based on MnO_x_ filter media often allow to treat groundwater without alkalizing or to reduce the alkali dose [34]. Typical filtration media for comprehensive groundwater treatment are: (i) Coated filter media among these can be distinguished media coated chemically most of all with MnO_x_ and media coated naturally by mixed iron and manganese oxyhydroxides, (ii) filter media consist of MnO_x_, in other words, the manganese ores of natural origin. 

The oxides used for divalent manganese removal have generally tri- and/or tetravalent manganese and “x” takes the value between 1.5 and 2.0 [35]. MnO_x_ has weak amphoteric properties and strong catalytic properties associated with the ability to donate oxygen [36]. It can occur in the amorphous or crystalline phase with various polymorphs [37,38]. The surface of MnO_x_ is acidic, in the solution it has a zero surface charge in the pH range of 1.5–5, which depends on the structure and age of the oxide [39]. A lot of researchers reported the ability of MnO_x_ to remove metals from water [2,40,41,42,43] in powdered form or immobilized on a carrier often of rocky origin. 

The study gives an overview of the characteristic of mineral materials of natural origin used as filter media in the groundwater treatment, which consist of MnO_x_ or are coated with MnO_x_ to improve their properties. With particular emphasis, this review presented the relationship between the properties of those amendments and the obtained technological effects, as well as the commonly used commercial products, their features and applications. The present work aims to provide new suggestions and useful information for further investigation of MnO_x_ consisting of and coated filter media for groundwater treatment.

## 2. Properties of Mineral Filter Media Commonly Used for Ground Water Treatment

### 2.1. Juxtaposition of Properties

Grain morphology, chemical composition, structural properties or density parameters of mineral media used for water treatment significantly determine their practical use. The chapter presents the main properties of commonly used rocks, such as quartz sand and gravel, chalcedonite, diatomite, glauconite, clinoptilolite, and anthracite, and describes the relationship of these features with the properties of filter beds in the general aspect of water treatment. The properties are shown in Table 1 and Figure 1. Detailed application of materials for groudwater treatment after enriching the surface with manganese oxides is discussed in Section 3.

### 2.2. Quartz Sand and Gravel

Sands and gravels are classified as sedimentary epiclastic rocks, whose skeleton has not been consolidated. The skeleton of sands may contain primarily all allogeneic components, e.g., quartz grains, feldspars, crystalline rock crumbs and/or sediment, mica and chlorite plaques and heavy minerals [54]. According to the varied mineral composition of sand a poli- or oligomineral character is possible. Also, multicomponent polymictic gravels and monomictic gravels built from one rock species can occur. High-quartz sands and gravels have practical significance in water treatment technology. Quartz is silicon dioxide, slightly accompanied by any admixtures of trace elements in the crystal lattice, resistant to various chemical substances, and in environmental conditions one of the most stable rock-forming minerals [47,51]. This determines the use of sands and gravels with a high content of quartz as filtration materials, whose impact on the chemical composition of the water is insignificant. A high content of quartz generates the materials to have high resistance to aggressive components of the environment, which ensures their long-term exploitation in filters, without the use of disintegration of rock grains. Quartz is characterized by low cleavage and significant hardness (Table 1). These features render into high mechanical resistance of quartz sands and gravel, which is a desirable feature for filter materials, because it determines their resistance during the operation of the filter bed.

In nature, crumb minerals with such good sorting of grains are extremely rare, as required by the parameters of filtration beds, therefore in industrially exploited rock deposits a raw material is used for obtaining fractionated aggregate. An important parameter of granular material is the degree of its sorting, defined in water technology as uniformity coefficient. The European Standard EN 12904: 2005 [72] precisely specifies the grains homogeneity of sands and gravels in commercial products whose uniformity coefficient should be less than 1.5. This parameter affects the geometry of the filter bed, which is a decisive feature of its hydraulic properties. The balance sheets of mineral deposits classify a group of natural aggregates as deposits of filtration sands and gravels. Quartz filtration media are also often obtained from foundry and glass sands. It is significant that gravel and sand are loose rocks that do not require crushing. Their grains are more rounded than crushed filter media. The sphericity of quartz sands is in the range of 0.8–0.9 [45]. Fine-grained sands have a more isometric shape and a sphericity closer to the unity compared to larger fractions [73]. This is due to the aforementioned mono-mineral composition of fine fractions. However, this is not a permanent feature of quartz sand, since according to the work of Beňo et al. [46] rounding of grains strictly depends on the origin of the rock too. In addition, it was specified that fine fraction of sand is more resistant to crushing [46]. The density parameters of quartz sand are presented in Table 1. Its bulk density is varied and depends on the grain’s size and compaction. These features are considered relevant for the hydraulics of filtration processes and the backwashing of mineral media in filters. According to the experimental data in Figure 1, flow velocity required to start expansion of quartz sand bed is average compared to other materials.

Requirements for sand and gravel as materials used for the production of drinking water were specified in European Standard EN 12904:2005 [72]. It stated that sand and gravel should have a crystalline structure, texture from smooth to rough, should be visually homogeneous and free of foreign matter. The standard admits that sand and gravel intended for drinking water treatment may be extracted in sand and gravel pits, in quarries by crushing or using dredgers. Because of the diverse origin of sand and gravel, it was assumed that their chemical composition may vary, therefore three classes depending on the content of basic components were determined. Type 1 sands and gravels should have the highest SiO_2_ content (minimum 96% by weight) and the lowest content of admixtures of sodium, potassium, calcium, iron, and aluminum. This type of sands should also contain a maximum of 2 wt.% substances that can be dissolved and washed away in an acid condition. Types 2 and 3 of sands and gravels can contain much less SiO_2_ (minimum 80 wt.%) and more admixtures. The high SiO_2_ content together with leaching resistance characterizes the resistance of sands and gravels to the damaging effects of the aquatic environment. Generally, quartz sands have the lowest specific surface area compared to other mineral filter media used in water treatment (Table 1), especially quartz sands with high mineral purity. That limits their use mainly as media for filtration.

### 2.3. Chalcedonite

Chalcedonite is a sedimentary, siliceous rock, the spiculite of biogenic origin [74]. The main building material of the rock is chalcedony, a fine crystalline and fibrous variety of quartz [75]. In the mineral composition of chalcedonite 68.3–95.4% vol. occupy chalcedony, opal, and autogenic quartz rocks [76]. It is solid rock extracted from deposit by open-cast method, and the output is subjected to crushing, fractionation, and drying. Because of the variability of deposits, chalcedonite filtration media can be characterized by two types of grains. Solid, sharp-edged grains with an even surface with singular slots have been documented, as well as more rounded grains with an extensive outer surface abundant in crevices and pores in which the remains of sponge needles were visible [49].

No European Standard has been developed yet for the use of chalcedonite as a material for the treatment of drinking water, however, chalcedonite from Poland has a hygienic certificate for this type of use [77]. In Poland, crushed chalcedonite is used in over fifty water treatment plants as filter media [78]. Commonly, it replaces quartz sand in filters, because of similar density parameters of these filter media (Table 1). It is beneficial since it does not require modernization of filter backwashing system devices. As shown in Figure 1, the expansion conditions of chalcedonite bed during backwashing with water are similar to quartz sand. Intraparticle porosity is a feature distinguishing both materials as filter media: of the chalcedonite filter medium is 0.48–0.52, while of the quartz sand is 0.41–0.43 [45]. This property contributes to the greater suspension mass capacity of the chalcedonite medium. Filter media with lower porosity are more effective in removing suspended particles from water, but they are faster to clog and increase hydraulic resistance, which shortens the filter run-time. Chalcedonite filter media have a specific surface area larger than that of quartz sand, yet definitely less than that of sorption materials (Table 1). Its hardness is slightly lower than the hardness of quartz (Table 1), which is beneficial for use in pressure filters.

### 2.4. Diatomite

Diatomite is a siliceous natural material also used in water treatment. It is a sedimentary rock of biogenic origin, with a biomorphic structure. The main component of the rock is amorphous opal, the least stable form of silica, however, the rock can also contain quartz, calcite, clay minerals, iron compounds, or glauconite [51,75,79]. As good sorbent, diatomite has many applications for removing various impurities from water [53,57,80,81,82,83]. The specific surface area of diatomite varies depending on the origin, though is larger comparing to the silica rocks discussed in this work (Table 1). The porous structure makes it an excellent filtration medium with a wide industrial application [84] and the porosity can be increased by modification [57,85]. Diatomite is also an excellent raw material for the production of powdered media for fine filtration. Powdered diatomite is used for clarification in filters in the industry, but also in water treatment technology. The properties of diatomite used in such applications are characterized in the European Standard EN 12913:2012 [86] for diatomaceous earth only. The standard specifies that in the product the content of silica must not be less than 85 wt.%, while the content of substances causing weight loss in 105 °C should not exceed 5% and the admixtures should not exceed 15%. No standard requirements were specified for the application of granular diatomite in drinking water treatment processes.

Diatomite is obtained by the opencast method, crushed and dried, additionally including the classification stage, because of the frequently encountered heterogeneity of the rock in the deposit. The density of diatomite is varied depending on its purity (Table 1), and for crushed grainy filter media it is higher than for powdered filter aids [52]. Compared to other mineral filtration media, diatomite has lower density, therefore less water is used to backwash diatomite filter beds than for other silica materials (quartz sand, chalcedonite) (Figure 1). In conventional filtration applications, lower hardness of diatomite may be an unfavorable feature due to the lower abrasion resistance of the filter media grains. 

### 2.5. Glauconite

Glauconite is a hydrated phyllosilicate of potassium, iron, and magnesium, which belongs to the mica group minerals [51]. The chemical composition of glauconite is dominated by silicon, iron, potassium, aluminum, and magnesium [58,60,62]. Glauconite occurs in the form of small grains with a diameter of 0.05–0.6 mm [59], sometimes 1.5–2 mm, forming glauconite sands or more solidified glauconite sandstones, containing from several dozen to even 90% of this mineral [54]. Glauconite grains take a variety of morphological forms, including rounded, oval, kidney-shaped, ellipsoidal or plate-shaped [87] with a flat or nodular surface [59]. 

Glauconite is characterized by the ability to exchange ions, resulting from its specific structure where the excessive negative charge generated in the crystal lattice is compensated by the exchangeable cations present in the spaces between the packets [88]. Cation exchange capacity of glauconite is varied depending on the origin 11–35 meq/100 g [58,60,89]. Glauconite can be chemically regenerated and in this form effectively removes metals from water, however, raw glauconite exhibits also exchange capacity. For this reason, glauconite can be used for water purification [58,60,90,91,92]. Its specific surface area is more extensive in comparison to the previously discussed silica materials (Table 1) and depends on the origin of the rock. If mineral material is used as the filter medium, density is important operating parameter. Density of glauconite is close to the density of quartz sand, which is why quartz filtration media can easily be replaced with glauconite. An example is Manganese Greensand, a popular material used for the treatment of underground water (described in Section 3.1.3).

The European Standard EN 12911:2013 presents the properties of manganese dioxide-modified glauconite, which can be used for the treatment of drinking water [93]. The standard specifies the typical chemical composition of glauconite, on the basis of which the filter material can be produced: SiO_2_ 52.6 wt.%, Fe_2_O_3_ 16.9%, Al_2_O_3_ 8.0%, K_2_O 7.0%, MgO 3.4%, FeO 3.2%, MnO_2_ 0.8%, ignition loss 7%. For reasons of water consumer health safety, a minimum content of toxic components in glauconite has also been determined: arsenic 25 mg/kg, cadmium 2 mg/kg, chromium 60 mg/kg, mercury 0.1 mg/kg, nickel 25 mg/kg, lead 5 mg/kg, antimony 5 mg/kg, and selenium 25 mg/kg.

### 2.6. Zeolite

Aluminosilicates with specific structural properties used in water purification processes are zeolites. The most common zeolites are analcime, chabazite, clinoptilolite, heulandite, laumonite, and phillipsite [54]. These are a porous type minerals [94] widely used as adsorbents, ion-exchangers, and molecular sieves [95], which contain uniformly sized pores in the range of 0.3–1 nm [96]. Zeolites are crystalline aluminosilicates of sodium, potassium, magnesium, and calcium, with negative charge on the structure balanced by exchangeable, mobile cations [97]. According to the statement, zeolites, depending on their origin, have a cation exchange capacity 64–229 meq/100 g [98]. These particular properties of zeolites determine their use for purifying water [65,83,95,97,98,99,100]. The ammonium ion removal on clinoptilolite is of a practical application, since it is characterized among zeolites by the highest selectivity for the exchange of this component [98]. Density of clinoptilolite is slightly smaller than the commonly used quartz filter media (Table 1) and the diversity of rock’s types results in the diversity of the bulk density. Characteristic for crushed materials, intraparticle porosity of clinoptilolite bed is 0.50–0.55 [45]. Variation also occurs in the specific surface area of zeolites and depends on the origin and type of zeolite and its content in the rock (Table 1). Zeolites are materials with lesser hardness and mechanical strength than quartz sand: clinoptilolite and phillipsite-analcime filter media can be used in various types of pressure and gravity filters, while less durable chabazite filter media is recommended only for gravity filters [64].

For water treatment natural zeolites are used, and only clinoptilolite, chabazite, phillipsite, and analcime are recommended for the production of water intended for human consumption, as defined in the European Standard EN 16070:2014 [64]. They ensure safe use without water contamination contained in the material by hazardous or toxic substances. The standard specifies the mineral composition of zeolites intended for the production of drinking water, together with the type of admixed minerals, because the mined rocks are rarely monomineral. For example, typical clinoptilolite zeolite used for water treatment should contain 85% (±5%) of this mineral, and the other ingredients may be cristobalite, feldspar, seladonite, calcite, quartz, clay, and glass. Chabazite zeolite should contain about 70% (±5%) of the main mineral, while the total content of both minerals in phillipsite–analcime zeolite must be greater than 50%. The standard clearly indicates that zeolites must not contain erionite, which is a carcinogen, included in the first group of carcinogens, developed by the International Agency for Research on Cancer [101]. For the safe use of zeolite materials in water treatment filters in the above mentioned standard also specifies the maximum content of harmful components, expressed in mg/kg of zeolite: arsenic 5 mg/kg, cadmium 2 mg/kg, chromium 30 mg/kg, mercury 0.1 mg/kg, nickel 30 mg/kg, lead 30 mg/kg, antimony 5 mg/kg, selenium 10 mg/kg [64]. The standard has recommended that each zeolite should have a cation exchange capacity of not less than 1.2 mg of ammonium ion per 1 g of zeolite.

### 2.7. Anthracite

Anthracite is a fossil solid fuel, classified as sedimentary organogenic rock. Among the fossil coal materials, anthracite has the highest carbon content [102]. Comparing to other filter media it has a low specific surface area, similar to chalcedonite (Table 1). Because of the low density (Table 1), anthracite is beneficial for usage as a filter material for water treatment. According to the data in Figure 1, flow velocity required to start the expansion of anthracite is lower than other materials. Anthracite is used in dual-media deep-bed filtration technique and clarifies the water much more efficiently and works more stable than quartz sand single-layer bed [68,103]. The upper anthracite layer in a multilayer bed provides pre-filtration of suspension which makes it possible to extend the filtration cycle [104]. Anthracite can also be a carrier for biofilm development in biosorption filters [105,106,107].

The parameters of anthracite as a filter medium for the treatment of water for human consumption are specified in the European Standard EN 12909:2012 [108]. Material for this purpose should have a carbon content above 90 wt.%. According to the guidelines of anthracite filter media, they should not react with water, especially acid, hence their solubility in hydrochloric acid is characterized, which should be less than 2% by weight. The anthracite mechanical strength allows it to be used in gravity and pressure filters. Because of the various uses and individual selection for multi-layer filters, anthracite is available in various granulations. For this reason, the recommended flow velocity and filter layer heights are only approximate and should be specified in detail in the individual multi-layer bed concept.

## 3. Manganese Oxides Coated Filter Media for Groundwater Treatment

### 3.1. Chemically Coated Filter Media

The manganese oxide coating of various filter media is a popular method in chemical as well as environmental engineering. Coating of a mineral carrier by manganese oxide with a given granulation changes the chemical and morphological properties of its surface, while allowing use in filtration systems. The particles of synthesized manganese oxides are small in size [2], therefore their use without carrier grains requires the use of fluidized-bed flow reactor [109] also improved by membrane separation [110].

#### 3.1.1. Coatings Characteristic

Production of a manganese oxide coating on a carrier is most often carried out by carrying out a chemical reaction in the presence of a solid phase of the carrier immersed in the reaction mixture. Finally the manganese oxide is chemically cross-linked to the surface of the carrier. Table 2 lists some commonly used procedures for coating of mineral materials by manganese oxides. The manganese oxide synthesis procedure significantly determines its polymorphic form, structure, presence of the crystalline phase. Various forms of manganese oxide are characterized by different effectiveness of contaminant removal [2], nanowires have almost five times greater maximal adsorption capacity of strontium and caesium than nanostructured microspheres [111]. The adsorption capacity of cobalt, nickel and zinc ions varies significantly among the different MnO_x_ with asbolane having the highest metal-removing potential, followed by birnessite and todorokite, and with manganite showing the lowest adsorption properties [112]. Feng et al. [113] have investigated the synthesized manganese oxides and proved that birenssite characterized by the largest surface variable negative charge has bigger adsorption capacity on heavy metals (lead, copper, cobalt, cadmium, and zinc) than cryptomelane, todorokite, or hausmannite. While carrying out research in the column tests it has been demonstrated that chalcedonite coated by birnessite type manganese oxide has more than three times greater adsorption capacity [114] than coated with amorphous manganese dioxide [115] (Table 2). Reaction conditions during coating such as pH and temperature, reagent introduction method, and reaction time also affect the result. As reported, the hydrothermal conditions indicate the transformation of Mn_3_O_4_ nanoparticles into γ-MnOOH nanowires, and the pH plays an important role in the transformation [116]. The example of quartz sand coating in reaction of permanganate with Na-lactate showed that an increase in pH in the range of 5–10 caused an increase in birnessite amount on quartz surface (from 0.14 to 0.26 mg Mn/g), as well as a portion addition of Na-lactate solution (from 0.26 to 0.58 mg Mn/g) and extension of contact time (in range 24–168 h from 0.56 to 1.29 mg Mn/g) [117]. Also the presence of additional anions can have an effect on the structure of manganese oxides. The sulphate ion-treated birnessite has the smallest nanofibrous crystals and the chloride ion-treated birnessite has the largest nanoplatelet crystals, as well as the dominant cryptomelane phase under sulfate treatment and mixed phase of cryptomelane and birnessite under chloride treatment was obtained [118].

In the coating process, an additional factor is the chemical composition of the mineral support, which can have a significant impact on the morphology of the produced coating. Manganese oxide phase resulting on the surface of high-silica clinoptilolite was characterized by nanorods about 10–20 nm in diameter and 500–700 nm in length, while coating on the low-silica clinoptilolite consisted of oval-shaped particles of 50–100 nm [119]. There were also differences in the thickness of the resulting coatings: very thin (12–50 μm) layer for high-silica carrier and deeply penetrating the grain structure, thicker (800–1100 μm) for low-silica clinoptilolite, because porous structure in low-silica samples allows the diffusion of the chemical inside [119]. The method of applying MnO_x_ to the carrier is also of great significance. Precipitation of MnO_2_ through the reaction (MnSO_4_ with KMnO_4_) in direct contact with the surface of siliceous carrier (chalcedonite) leads to the formation of a more durable coating than when the precipitated MnO_2_ is thermally (100 °C) bounded to the carrier [120]. The chemically bonded coating is more durable and does not peel off like the thermally bonded coating, although it is thinner.

Other work also indicates that mineral carriers are not always completely covered with MnO_x_, which are patches distributed on the surface and at least 96% of the produced birnessite did not adhere to the quartz surface [117]. This is particularly important when producing MnO_x_ coated filter media because the coatings may break off during transport, and above all during stage of backwashing in the filter. The work of other authors indicates the production of coated quartz sand, diatomite and zeolite, on the surface of which MnO_x_ is evenly distributed [41,119,121,127]. What is especially important for minerals with Mohs hardness less than quartz, coating their surface leads to an increase in their mechanical strength [41,119]. As shown in Figure 2, both ways prepared coatings were occupied by newborn manganese oxides, which formed a globular shape. The globules formed in solution and thermally bonded are finer (diameter about 0.25 μm) (Figure 2a) and the globules formed in direct reaction with the support surface are twice as large (diameter about 0.5 μm) (Figure 2b) [120]. Morphologically diverse forms of MnO_x_ can be obtained depending on the mixing during synthesis: black nanofibers birnessite under mechanical stirring during an initial stage of synthesis was obtain, whereas granular particulates of brown birnessite without stirring [129]. This condition can be particularly important when coating large grains of mineral media with high density, when it is difficult to ensure adequate mixing.

Coating of a mineral support leads to the production of a composite material whose specific surface area is developed [131]. It has been experimentally shown that the increase in content of birnessite layer on the quartz support dominantly contributed growth its specific surface area [117]. According to the data collected in Table 2, the increase in specific surface area varies, from minimal changes in quartz sand to more significant in chalcedonite. In the case of diatomite, the data are ambiguous, because the size of the specific surface of the carrier with the coating is also affected by the method of its synthesis. This confirms the huge increase in the diatomite specific surface (from 30 to 178 m^2^/g), on which the nanowires coating was produced using the hydrothermal method [132].

#### 3.1.2. Adsorption Properties

The values for maximum adsorption capacity of coated mineral media are also different (Table 2). Definitely the coated diatomite and zeolite have a higher adsorption capacity for metals than coated quartz sand and chalcedonite. In the main, MnO_x_ coating of siliceous carriers causes a significant increase in adsorption capacity of modified materials comparing to the raw: for coated chalcedonite from 0.084 to 0.918 mg Mn/g [133] or 1.07 mg Mn/g [124], for diatomite from 8.5 to 56.8 mg Pb/g [127] and from 24 to 99 mg Pb/g [126] as well as from 51.6 to 76.3 mg Cr/g and from 17.8 to 61.7 mg As/g [132]. Similarly, zeolites as a result of modification can increase adsorption capacity from 0.259 to 1.123 meq Mn/g (respectively 7.11 and 30.85 mg Mn/g) [41]. Other studies indicate that surface MnO_x_ coating on zeolite can lead to both an increase (from 9.1 to 21.6 mg Mn/g) as well as a decrease (from 18.2 to 11.2 mg Mn/g) of its adsorption capacity, which is explained by the fact that a part of the cation exchange complexes in the zeolite is excluded from the adsorption process since manganese oxide forms in them [119]. Kinetics studies of heavy metals and metalloids adsorption on diverse MnO_x_ coated mineral media (quartz sand, chalcedonite, diatomite, zeolite) confirm that process follows a pseudo-second-order model and is chemisorption [41,122,125,134,135,136]. In the analyzed systems adsorption was endothermic in nature and occurred spontaneously.

Experimental conditions are another factor affecting the diversity of adsorption capacity. However, the operating conditions like the solution chemistry (pH, ionic strength and ion types), temperature, experimental form (batch or column runs) are quite different [2]. It has been shown that the adsorption capacity of amorphous manganese dioxide-coated chalcedonite obtained in column run is over twice as large as in batch (3.11 and 1.25 mg Mn/g respectively) [137]. In addition, it should be taken into account that in the case of removal of manganese divalent cations on MnO_x_, it does not have to be limited to adsorption or ion exchange, because a redox reaction is possible between the components. The MnO_x_ skeleton is dominated by tetravalent manganese cations, some of which are replaced by divalent or trivalent cations, giving it an excess of negative charge, therefore it can initiate redox reactions or embed other cations into the intra-network structure [138,139]. The experimentally determined maximum adsorption capacity depends on the conditions that affect the interaction between MnO_x_ and solution components, hence the importance of isoelectric point of MnO_x_ as well as pH and ionic strength of adsorptive solution is analyzed. At pH lower than pH of isoelectric point, removal of metal ions from solution is inhibited as result of a competition with hydrogen and as the pH increased, the negative charge density on MnO_x_ surface increases due to deprotonation of the metal binding sites and thus the adsorption of metal ions increased [40,122]. Not only the variable surface charge of MnO_x_ affects the adsorption capacity, but also a significant role is played by the hydration tendency of the adsorbed metal [113], as well as its cation or oxyanion formation [5]. The question is more complex, because of the possible occurrence of some adsorption reactions without the release of proton, irrespective of ionic strength and the nature of electrolyte solution, which suggests the metal adsorbs on MnO_x_ with high surface charge not only as free ions, but also as some of its complexes [140]. In addition, the removal efficiency of metals is intensified because of precipitation of hydroxides at higher concentration of component and at higher pH [40] as well as because of higher co-anion valence leads to a higher adsorption capacity [141].

High variability of research conditions determines the effects and thus a direct comparison of the reported data seems inappreciable [5]. As a solution, a procedure could be introduced, standardizing the measurement of the adsorption capacity of manganese oxides-coated media. Since batch tests are easy and very popular, the standardized conditions could include this kind of experiment parallel to the column runs. The flow conditions are recommended for measuring the oxidation capacity (understood as dissolved manganese uptake capability) of manganese greensand or pyrolusite described in European Standards EN 12911:2013 [93] and EN 13752:2012 [142] respectively. Similarly, the report of Tobiason et al. [143] attempts to standardize the procedure of measuring adsorption capacity of media coated in filters treating the groundwater. Flow conditions were also applied. Table 3 summarizes the specific conditions of these procedures. As can be seen, proposed conditions also vary (flow velocity, media mass, manganese concentration). What is crucial in the standards [93,142] and the report [143], the test water is enriched with bicarbonates responsible for alkalinity. This is of great importance while conveying the results of laboratory tests to technological conditions. In this context, there is a need to use standardized procedures for batch tests, but also for removing other metals from water. This is relevant for better use of published research results. Comparing adsorption capacities of coated filter media is also difficult because, although it significantly depends on the amount of oxide on the carrier grain and the size of the specific surface area, it is most often expressed in terms of the unit weight of the whole composite material. In chemically coated filter media, the carrier dominates quantitatively over the coating and it has been shown that for this type of materials it is better to compare adsorption capacity in terms of unitary value of specific surface area [144]. Bruins et al. [145] considered this need when comparing filter media of varying density and proposed a reference to the unitary volume of material.

As shown in the data presented in Table 2, the amount of MnO_x_ deposited on a mineral support is varied and most often ranges from a few tenths to several mg/g of coated filter media. The amount of coating can be increased through subsequent coating cycles, which results in an increase in adsorption capacity. This was shown on the example of manganese dioxide-coated chalcedonite, which as a raw material has a marginal adsorption capacity of 0.084 mg Mn/g, and after subsequent coating cycles (1–4) the capacity increases to 0.918, 1.711, 2.024, and 2.217 mg Mn/g respectively [133]. The use of double coatings was found to be the most advantageous, because the adsorption capacity increases 20-fold relative to the silica support, and further modifications no longer bring such significant increase. Visual changes occurring on the silica support after subsequent coating cycles are shown in Figure 3. Another phenomenon noted is the change in the surface structure of MnO_x_ after metal adsorption from solution. After adsorption of lead ions on the MnO_x_-coated sand its specific surface area is decreasing from 0.712 to 0.552 m^2^/g, which suggests that the part of pores were occupied with lead ions because of average pore diameter decreases simultaneously [122]. Additionally, the acidic regeneration allows to remove lead ions with return to baseline properties. Completely unalike relationships were noted during the removal of manganese on MnO_x_-coated mineral media, forming the Mn^2+^/Mn^4+^ system. Specific surface area of manganese oxide-coated chalcedonite increases after manganese ions adsorption from 9.88 to 12.09 m^2^/g which is accompanied by increasing of meso- and micropore volume from 0.030 to 0.035 cm^3^/g [130]. After adsorption of manganese, the surface of the MnO_x_ on chalcedonite changed its appearance and chemistry assessed by SEM and XPS techniques: globular clusters containing MnO_2_ (BE 643.45 eV) and Mn_3_O_4_ (BE 641.73 eV) (shown in Figure 2b) were covered by web of MnO_2_ (BE 643.45 eV) and MnO (BE 641.06 eV) resulting from oxidation of manganese removed from water (Figure 4) [130]. This is confirmed by the results of studies on MnO_x_ coated sand, the specific surface area increased from 1.99 to 2.19 m^2^/g after adsorption of manganese [121]. Investigation of surface of MnO_x_-coated zeolite before and after manganese ions removal shows that all adsorbed manganese is in the oxidized form i.e., the MnO_x_ in coating effectively oxidizes divalent manganese from solution into tri- and tetravalent [41]. Many authors suggest the divalent metal species, also manganese, are removed by a combination of several interfacial reactions namely ion-exchange, chemisorption, and adsorption [40,41,123]. MnO_x_ containing trivalent and/or tetravalent manganese react with adsorbed divalent manganese in redox reaction. Finally, when sorbed manganese is oxidized and manganese from the coating is reduced, the removal capacity of coated media is exhausted [35]. The capacity is renewed through chemical regeneration using an oxidant, which is a characteristic feature of almost all MnO_x_-coated filter media formed in chemical formulation used to remove manganese from groundwater.

#### 3.1.3. Commercial Media Characteristic

In this section, commercial materials of this type are discussed in terms of technological conditions for groundwater treatment, and their features are listed in Table 4. They have the form of small, spherical grains of dark brown color. Manganese Greensand was the first original commercial deposit for the treatment of groundwater, made by the chemical formulation of glauconite. Currently, the producer, because of too high demand in relation to the possibility of mining glauconite sand, replaced it with a material called GreensandPlus^TM^, which is produced on a silica carrier [146]. Other filter media, MTM^®^ and BIRM^®^, also contain siliceous carrier. Producers in catalogue sheets do not provide information on the composition of the coating and carrier and their mass proportion, such data can be taken from the material safety data sheets (shown in Table 4). Manufactured coatings are characterized as manganese dioxide or manganese oxide in one case, but the producers do not specify their phase analysis in detail, which as described in Section 3.1.1 may affect removal efficiency of water compounds.

Commercial materials, distributed worldwide as filter media for groundwater treatment and having the word “greensand” in their name, were analyzed. Their significant differences in compositional, structural, and physical attributes were proven in work of Outram et al. [155]. Analysis of the mineral composition showed that two out of five have a quartz carrier, while none of the “greensands” contained glauconite. The carrier in other three materials was the mixture of manganese dioxide and montmorillonite as well as quartz and hematite. The coatings commonly called manganese dioxide was also investigated, and in their phase composition of pyrolusite, ramsdellite, romanechite, and cryptomelane were detected [155].

Analyzing the share of carrier and coating in commercial materials from Table 4, it can be assessed that Manganese Greensand, GreensandPlus^TM^ and MTM^®^ are similar in this respect, with the chemically active coating being the smallest on MTM^®^ material, and practically the same from those from one manufacturer on Manganese Greensand, GreensandPlus^TM^. In this case, BIRM^®^ looks significantly different, in which the share of the coating is many times greater. This translates into operating conditions, since three filter media with a small amount of coating are characterized by removal capacities understood as the maximum mass of reduced water constituents that can be oxidized by a unit volume of the filter bed. In redox reactions with MnO_x_ coating the removal capacity is exhausted and requires renewal with the regeneration oxidizing agent, like chlorine or potassium permanganate. These materials can be named chemically regenerated filter media. Another type is BIRM^®^ material, which contains several times more MnO_x_ and does not require regeneration, as well as no mass limit in the form of the removal capacity parameter. This allows it to be classified as catalytic filter media, with a similar mechanism to the manganese ore filter media described in Section 4.

#### 3.1.4. Operating Conditions and Examples

MTM^®^, GreensandPlus^TM^ and Manganese Greensand filter media enable the removal of dissolved iron, manganese, and hydrogen sulphide from groundwater, and the producer of the GreensandPlus^TM^ additionally indicates its use for removing arsenic and radium. Arsenic and radium compounds are removed in the presence of iron and manganese respectively throughout the oxidation and co-precipitation [149]. An important technological aspect is that on MTM^®^, GreensandPlus^TM^ and Manganese Greensand it is possible to carry out the water treatment process already at pH 6.2, which is very important in the case of groundwater, which is often slightly acidic. The flow velocity and bed depth used in the filter depend on individual technological solutions. The values given in Table 4, although slightly different, are typical for groundwater treatment plants. All materials can be used as fillings in various structures gravity fed and pressurized filters. It should be added that GreensandPlus^TM^ with a silica carrier has higher mechanical strength and is more resistant in pressure filtration applications than Manganese Greensand based on lower strength glauconite. Backwashing filter beds with water is used after the filtration cycle to remove retained suspensions and part of products of oxidation. The conditions of backwashing with water for MTM^®^, GreensandPlus^TM^ and Manganese Greensand are similar and at reverse flow velocity 24–30 m/h consist in causing the expansion of 20–40%, 35–40%, and 40% respectively. 

Granops [156] describes the modernization of groundwater treatment plants, in which the use of Manganese Greensand significantly improved the manganese removal efficiency from both soft and hard water. At high iron content (~5 mg/L), single-stage filtration on this type of bed is insufficient, as the fine granulation of the material hinders its operation under conditions of the formation the big amount of iron hydroxide clogging the bed. This leads to increased hydraulic resistances which requires more frequent backwashing and implies shortening of the filtration cycle. Therefore, in technological solutions, the author consistently proposes to use over Manganese Greensand bed an anthracite layer with appropriately selected granulation to retain oxidized iron suspensions [156]. The result of research on the treatment of groundwater in the aeration + filtration technological line confirm the high efficiency of Manganese Greensand and justifies that because of the formation of excessive resistance on a bed with low grain granulation, it should be used in second-stage filters [157]. These solutions are different from the simplest one, according to which Manganese Greensand removes manganese and moderate amount of dissolved iron through chemical oxidation without water pre-aeration, only with regeneration by an oxidant and with backwashing [6].

The usage of BIRM^®^ for the treatment of groundwater is beneficial since it does not require the use of chemicals for regeneration and has low density and bulk density, making its backwashing cost-effective. It is also characterized by technological restrictions, as it is only intended for removing dissolved iron and manganese. Water to be treated must be heavily oxygenated—dissolved oxygen content at 15% of iron and 29% of manganese content at pH ≥ 6.8 [153]. An additional requirement is that hydrogen sulfide, polyphosphates, oil, as well as organic matter TOC > 4–5 mg/L and free chlorine >0.5 mg/L may not be present in the water, as they can destroy the catalytic coating [153]. Barlokova’s research shows that BIRM^®^ is very sensitive to inappropriate operating conditions and may lose properties, in this case insufficient water aeration, because the dissolved oxygen content in water of 15% of the total iron and manganese content resulted in exceeding the concentration of manganese in the filtrate, and the material required activation by regeneration with potassium permanganate [63,158]. Additional operational restrictions arise from the need to ensure that bicarbonates and hydrocarbonates (total alkalinity) have a double advantage over the sum of chlorides and sulfates [153].

Other less common commercial products are coated zeolites Klinopur-Mn and Klinomangan. The result of technological research at experimental stands at the water treatment plant was the conclusion that Klinopur-Mn removes iron and manganese from the water with the same efficiency as BIRM^®^ and Manganese Greensand [63,159]. Klinopur-Mn is produced from zeolite mined in eastern Slovakia from the Nižný Hrabovec deposit, composed of clinoptilolite (84%), and the other ingredients are cristobalite (8%), feldspar (3–4%), and illite (4%) [158]. The mineral carrier of Klinomangan came from the Rátka deposit in Hungary and contained less clinoptilolite (55%) and more other ingredients such as cristobalite (15%), feldspar (10%), and montmorillonite (10%) [160]. After chemical coating Klinopur-Mn and Klinomangan contained 6.92 and 15.16% of MnO_2_ respectively [160]. In the research conducted in the technological system at water treatment plant, it was shown that only the presence of the manganese oxide coating was important in the process of removing manganese from water, because both zeolites, despite the different composition of the carrier, achieved the same efficiency of treatment. Manganese dioxide-modified chalcedonite is similarly effective in the conditions of on-site water treatment plant as a rapid filter bed, which effectively removes iron and manganese from quaternary groundwater, achieving removal capacity of 0.67 g Mn/L [161], the same as MTM^®^ and GreensandPlus^TM^. Interestingly, in independently conducted studies on the removal of manganese from groundwater by coated chalcedonite and Klinopur-Mn, it was noted that in the following filtration cycles (separated by chemical regeneration potassium permanganate and backwashing) there is an increase in run time and removal capacities of filter media [115,158]. The authors speculate that this is the result of extending the structure of the coating and increasing its specific surface area. They also speculate that after some time no regeneration will be required, as in the filters with naturally coated media.

### 3.2. Naturally Coated Filter Media

#### 3.2.1. Coating Formation

Treatment of groundwater by aeration and rapid filtration method including biological oxidation is widespread in Europe [29,31,162,163] and in the world [164,165,166,167,168]. At treatment plants, this method is used in the systems based on open or closed aeration and one, two or more filtration stages. In this method spontaneous formation of coating occurs on the filter medium grains. Many authors report the use of naturally coated filter media for simultaneous removal of iron and manganese [164,169,170] as well as iron, ammonium ion, and manganese [162,171,172,173,174]. In effect active oxidation zones are created in the filter bed profile. Referring to side of the filter supplied with aerated raw water, in the zones there are catalytic and bio-catalytic transformations of iron and manganese or iron, ammonia, and manganese, respectively. Based on the review of scientific articles it has been determined, that in this type of systems the removal of iron and manganese can take place based on biotic processes or their combination with physicochemical processes, whereas the ammonium ions are oxidized mainly by microorganisms [31]. Unlike aerobic nitrifying bacteria, manganese and iron oxidizing bacteria belong to microaerophiles, and therefore their optimal development occurs under conditions of limited oxygen dissolved in water, as well as they are classified as organisms developing in a transition environment. Some studies point to predominantly biotic way of manganese and ammonium nitrogen oxidation [175] and abiotic oxidation of iron [166]. Similarly was confirmed that biotic processes play a key role in dissolved manganese removal because the abiotic oxidation on MnO_x_ surface is too slow [168]. Also it has been proven that during the start-up period the oxidation is consistently of biological origin and over a prolonged filter run time of mature bed the removal based on physicochemical interactions [176] especially on adsorption and heterogeneous autocatalysis [167]. Biological treatment is possible even on ripened filter media and under slightly acidic conditions [177]. Quite dissimilar conclusions were drawn from the experiment of groundwater treatment on iron-manganese oxides-coated filter media; the important role the ammonium ions and manganese transferring play in the abiotic catalytic oxidation [178]. Some authors report additional oxidation and removal on coated media of arsenic [165,166,168,179,180], phosphorus [168,181], and antimony [165], although it is rather abiotic. Groundwater composition, its Eh, pH, temperature, and oxygen concentration, but also flow velocity, media type, and backwashing method are the main factors for treatment of aerated water by filtration [31] and can determine the nature of the process. These threads link to the conclusion, that it is very difficult to generalize which processes (biotic or abiotic) predominate in naturally coated filters and may result from individual specific operating conditions.

Quartz sand or gravel are the most popular materials used in field operations based on natural coating [29,165,166,176,178,182,183,184]. Among the less frequently used mineral filter media clinoptilolite [185], anthracite [167,183], or chalcedonite [128,186,187] can be mentioned. However, during the filtration of aerated water, the grains of filter media are gradually covered with iron and manganese oxyhydroxides. In the start-up period, the filter slowly achieves higher and higher treatment efficiency until satisfactory results are obtained. The duration of this period varies and depends on many factors, such as chemistry of groundwater, technological conditions of water filtration, filter design, and properties of the filter medium, implies a rate of ripening [31,145]. A summary of information on this topic is provided in Table 5. Duration of start-up period in many cases is brief (about 20 days) but can be also prolonged (many months), especially when chlorinated water is incorrectly used for backwashing, as in the last two examples. Systematic introduction of a chemical oxidant into the filter together with the backwashing water can be a factor that inhibits the growth of microorganisms. By some means, this observation can confirm the participation of microorganisms in the activation stage. Based on accounts of engineering professionals has been generalized that typically start-up takes more than 2–3 months, but in some exceptional cases even more than a year [176]. Analyzing the compiled data (Table 5) it is difficult to determine the relationship between start-up time and filter media type, because of the large diversity of other operating parameters. Some studies prove that silica filter media (quartz sand and chalcedony) of different structure and genesis needs the same start-up time for effective manganese removal (20–21 days), but at the same operating conditions Paleozoic basaltic rock characterized by a larger specific surface area and different mineral composition was ripened slightly faster (15 days) [128]. It was documented, that the specific surface area plays a significant role in biotic oxidation of manganese [188]. Studies on the ripening of virgin sand and sand/anthracite beds showed that the start-up period lasted the same time and was not dependent on the type of filter medium, rather on operating conditions like backwashing frequency and filtration flow velocity [189].

#### 3.2.2. Coatings Characteristic

During water treatment, the naturally formed coating undergoes constant layer-by-layer growth and its thickness is determined by surface abrasion during filter backwashing. According to various authors, the final coating after many years is 50–90 vol.% [192] and 63.7% [193] of mature filter grain. It should be noted that some authors reported the lower mass of naturally surface coating of 31.7 ± 2.9 mg/g and 78.3 ± 10.4 mg/g after 3 and 15 years of ripening respectively [167], which corresponds to about 3 and 8 wt.%. The amount of manganese in coating on filter media can be very diverse: from 0.01 mg Mn/g media to more than 100 mg Mn/g media [143], yet it is demanding to compare it with natural coated media because of the chlorine used before the filters. The report [143] reviews that large variation in the vertical profile of the filter bed was observed and regardless of the type of media, the upper layer of the bed contained more manganese in the coating. Commonly, naturally coated filter media have a higher coating content than chemically coated filter media described in Table 2 (average less than 1 wt.%), as well as some commercial chemically coated filter media described in Table 4 (1–3 wt.%). The greater thickness of naturally formed coatings causes a notable increase in their specific surface area compared to the chemically formulated ones. It is likely that coatings on chemically coated media will also grow during the lifetime of the filter, although the authors cannot cite the confirmation. From practice it is known that excessive growth of the coating on grains is observed with improper filter operation like too low backwashing intensity. 

The small specific surface area of quartz sand (carrier) after ripening process may increase from 0.064 m^2^/g to 181.5 m^2^/g [194] or to 110.5 m^2^/g [193] of media from manganese removal zones. The produced coating is highly porous, and the total pore volume of quartz media before and after ripening can increase from 0.0002 cm^3^/g to 0.343 cm^3^/g [194]. The increase of specific surface area of coatings obtained by the natural method can be very large, almost a hundred times larger compared to chemically coated quartz sand media, as well as coated media of other carriers (Table 2). Other sources indicate that the specific surface area of naturally coated sand formed for many years at groundwater treatment plant may be smaller and amount to 39.6 m^2^/g and 8.3 m^2^/g [144] for media from iron and manganese removal zones respectively. Media from top of filter bed removing iron, ammonium nitrogen and manganese after 3-year exploitation have increased their surface to 9.1 m^2^/g [179]. Already during the short 12-day start-up period, the quartz sand specific surface area increased from 0.106 to 3.920 m^2^/g [182]. The size of the specific surface area of matured media depends on the thickness of the coating, which in turn depends on the operating conditions of the filter bed. It should be mentioned that engineering practice shows that excessive growth of grains of filter sand causes technological problems in the filtration system and, as a result, deterioration of the quality of treated water.

Naturally coated filter media are characterized by a significant heterogeneity of the surface compared to chemically coated. X-ray mapping analysis demonstrates the heterogeneous distribution of iron and manganese elements on the surface of naturally coated anthracite [167]. Media of both iron and manganese removal zones from groundwater treatment filters contain simultaneously iron and manganese [144,194]. The elemental composition of formed coatings identified at various groundwater treatment plants is presented in Table 6. Iron and manganese, as well as oxygen from their oxyhydroxides, are the major elements. The surface of coated anthracite also contained carbon coming from biofilm or carrier [167]. The analytical method used has noticeable impact. The authors determined by extraction that 3-years old filter media contained 3.4 ± 0.5 mg Fe/g and 5.0 ± 0.3 mg Mn/g, while further ripening led to increase to 12.9 ± 0.6 mg Fe/g and 19.9 ± 3.3 mg Mn/g on 15-years old filter media [167]. Of the side elements, calcium and silica are the most common in coating, sometimes magnesium, phosphorus, and aluminum, while sodium, potassium, and sulfur are definitely less present (Table 6). The occurrence of side elements in the coating depends on the composition of the treated water and the susceptibility of its components to deposition in the form of solid into the coating. This applies mainly to the precipitation of calcium, silicon, magnesium, aluminum, or oxidation of hydrogen sulfide to sulfur (next to the main oxidation of iron and manganese to oxyhydroxides), which occur under the influence of pH and Eh changes as well as changes in calcium carbonate equilibrium in aerated and filtered water. Jones et al. [195,196] have proved the presence of aluminum in coatings to be the result of capturing the soluble and insoluble species and that the deposition of aluminum into coating is concurrent with MnO_x_ accumulation. Using the XPS technique it has been shown that naturally occurring coating consists of iron and manganese as well as the organic material of microbial origin [197]. As has been specified, the 3- and 15-years filter media from groundwater treatment plant contains dry microbial mass of 3.5 and 1.4% of the total coating weight respectively [167]. Using a different test method, it was determined that the total organic carbon content in naturally formed coating was below 0.1 wt.% [166]. The spatial distribution of microbial populations along with the depth of the filter revealed the stratification of iron, ammonium, and manganese ions removal [166,174] therefore it should be emphasized that in the filter bed profile the composition of the coating is varied. A entirely dissimilar property is the homogeneity of sand surface after 20-day ripening obtained by EDS mapping. As the authors emphasize, it does not coincide with literature reports [182] and it must have been due to the specific conditions of formation.

Mature coatings forming on naturally coated filter media are identified as poorly crystalline [183,194] and sometimes as entirely amorphous [197]. Contrarily, a large number of crystalline phases on the coatings of media from the iron and manganese removal filters were detected, what the authors explain by the fact that more than 10 years of ripening of the media resulted in the crystallization of iron and manganese oxyhydroxides [144]. Other authors have identified small and dense crystalline structures in the coatings, mixed with less dense, poorly crystalline or amorphous manganese oxyhydroxides [195]. In coatings of filter media from operating groundwater treatment plants birnessite type MnO_x_ was detected [176,183], while poorly crystalline nature is typical of this MnO_x_ variety [112]. Jones et al. [195] also reported the presence of birnessite and manganite in coatings formed on filter media. Using the EPR technique, it was determined that birnessite is of biological origin during the start-up period, and with a prolonged filter run time biomineral form occurs while at the end in ripened coatings birnessite of physicochemical origin dominates [176]. Katsoyiannis and Zouboulis [197] reported that as a result of groundwater treatment the biogenic, amorphous tri- and tetravalent manganese oxide and also iron oxide in ferrihydrite form on the coating of filter media was formed. This requires attention because biogenic manganese oxides are characterized by higher adsorption capacity than oxides of abiotic origin (e.g., birnessite) [198]. In another research also birnessite was identified in the coating from the manganese removal filter, while iron oxides, hematite and maghemite, on the iron removal media were also confirmed to be present [194]. The intermediate valence of manganese (from +3.5 to +3.9 [199]) in the birnessite can imply the high reactivity of these type of coatings and the special ability to catalyze the oxidation of manganese during groundwater treatment. 

It should be emphasized that the heterogeneity of the chemical and phase composition as well as the morphological diversity of the naturally formed coatings make it very difficult to compare it with chemically formed, obtained in laboratory, or in factory conditions. The dual nature of biotic and abiotic processes occurring on naturally coated filter media is an additional aspect that distinguishes them from chemically coated ones. Because of the regeneration with oxidants, the commercial coated media purify water mainly based on abiotic way.

## 4. Manganese Oxides Consisted of Filter Media for Groundwater Treatment

### 4.1. Manganese Ores Properties

Manganese rocks of sedimentary, volcanic-sedimentary, and in many cases metamorphic origin are represented by carbonate, oxides, and hydroxides of this element, however, deposits with high manganese content, so called, manganese ores, formed as a result of hypergenic processes in the weathering zone, have fundamental commercial significance [200]. Those are made of minerals in colloidal or small-crystalline aggregates form, with tarry appearance and black or brown color. The most popular is pyrolusite with less common: ramsdellite, psilomelane, manganite, and nsutite [102,200]. Manganese ores are exploited by opencast method as well as a deep-mining one. Output is crushed, fractionated, and unified by mixing different batches of material. The chemical composition of ores is similar, e.g., they all have a high manganese content—over 80 wt.% in conversion to MnO_2_ and over 50 wt.% in conversion to Mn. In addition, they contain admixtures of other elements, such as iron, silicon, aluminum, and calcium [201].

Even if the chemical composition of manganese ores is similar, their phase composition is highly diverse and generally the manganese compounds occur in several different crystalline phases. As presented in work of Sorensen et al. [201] the output from the Wessels mine (RSA) contains manganese in bixbyite form, braunite, manganite, and hausmannite; the Groote Eylandt ore (Australia) can be distinguished by more homogeneous phase composition and contains primarily pyrolusite and in less quantity cryptomelane; in CVRD material (Brazil) semicrystalline nsutite was dominating and in sequence todorokite and pyrolusite; the phase composition of the Gabon ore contains above all cryptomelane and nsutite, then pyrolusite and a trace amount of manganese in ramsdellite phase. The diversity of manganese ores can be high, as evidenced by the results of the phase analysis of Gabon manganese ore for water treatment, in which manganese was identified only in pyrolusite phase and its polymorphic form Mn_0,98_O_2_ [194]. It also happens that manganese ore does not occur only in the oxide form. The ore with a high content of manganese (78.8% Mn) from Indonesian Sumbawa can serve as an example, in which in addition to the main oxide form (pyrolusite) it also contains rhodochrosite (carbonates) and polymorphic rhodonite variation (silicates) [202].

### 4.2. Commercial Media Characteristic

After crushing and fractionation, ores are used as a catalytic filter material for removal of mainly manganese from water as well as iron, sulphides, and arsenic [33]. Manganese oxides have the adsorption properties and play the role of a catalyst leading to oxidation of manganese and other reduced components and retaining them in the filter bed. In contrast to naturally coated filter media the manganese ores media does not need the start-up time for effective water treatment. Even if biofilm develops on grains in favorable environment, biological oxidation is not the core of operation of these materials. In comparison to chemically coated filter media, the manganese ore can treat groundwater by catalytic oxidation without regeneration. In the groundwater treatment technology ores containing MnO_2_ have practical significance whose concentration in the deposit ranges between 70 and 80% [32]. Requirements for the quality of manganese ore as a material used for the treatment of drinking water were specified in European Standard EN 13752:2012 [142], and the average chemical composition of manganese ore is determined for two categories of mineral resources. Ores from the first class of catalytic filter beds are characterized by a higher manganese content (80–90 wt.% of MnO_2_) and smaller amount of non-manganese admixtures (iron, silica, calcium, magnesium and aluminum compounds). In the standard is clearly states that not every variety of MnO_2_ is suitable for use as a catalytic bed, however, the catalytic activity of manganese ore varies notably depending on the origin and is not related to the content of manganese in the deposit. Manganese ores diverge also in hardness. The first class includes ore resistant to abrasion, with a Mohs hardness of 6 or more, and commercial products of this class as a result of backwashing filters lose on average 3% of weight per year and are considered as materials not deteriorating in technological process of water treatment [142]. They can be used as independent filter fillings or as one of the layers in the filter (usually the bottom one). The second class includes manganese ores with lower MnO_2_ content and hardness less than 6, which is associated with grater abrasion of grains and weight loss of the deposit. Therefore, those are referred to as consumable products that can be used in filters as first class ores or mixed with quartz sand, without creating separate layers.

Manganese ores are distributed in form of catalytic filter media with various trade names. In the United States, South America, and Australia they are known as Pyrolox, Filox-R, and MetalEase brands. In Europe they are available as G-1, Multiman 3M, or Defeman. In Table 7 basic parameters of these are presented, determined on the basis of data provided in catalogue sheets. Manganese ores for water treatment have high and alike MnO_2_ content. Inconveniently, producers refuse to pass over the phase composition and origin of mineral resource. Only the producer of G-1 media states that the material comes from Moanda mine in Gabon and contains pyrolusite [203]. It is known from other sources that Pyrolox contains ramsdellite, nsutite, and pyrolusite [204]. The specific surface area is not large: G-1 16.5 m^2^/g [144], Gabon manganese ore 7.25 m^2^/g [194], Pyrolox 2.58 m^2^/g [204].

### 4.3. Operating Conditions

The recommended optimal pH range of treated water is differential (Table 7) and related to the used treatment technique. With neutral and slightly alkaline pH water treatment consists of aeration and water filtration. The usage of manganese ores for water treatment with slightly acidic reaction (allowed for MetalEase [205]) may require dossing the oxidizing agent (chlorine, ozone, potassium permanganate) to water without the need to aerate it. The technique is commonly used since the usage of oxidants leads to by-products creation that worsens water quality [35]. Producers propose a wide range of rates of water filtration through beds of manganese ores media: lower values are predominantly recommended for solutions with single-stage filters and higher at two levels of filtration. The suggested thickness of the filtration layer is associated with providing the required time of contact of water with the surface of filter media, providing the full course of the catalytic oxidation reaction. Taking data from the Table 7 regarding flow velocity and depth of bed, it was determined, that the contact time should range from 1.5 to 3 min. Commercial manganese ores media exploited in the presence of chlorine and at pH < 6.2 needs contact time less than 1 min to achieve 100% manganese removal [109]. The adsorption capacity for manganese ions is not specified for manganese ore filter media because their properties are not depleted. Producers, however, specify the maximum concentration of iron, manganese, and hydrogen sulphide in the water flowing into the filter, thus indicating the threshold for the effective operation of catalytic media. In addition to the high efficiency of water treatment, catalytic media also provide operational advantage resulting from extended filtration cycles and low increase in hydraulic resistance, since the coatings formed on their surface, created in the catalytic oxidation process, have more compacted structure than the sediments retained on the quartz bed.

The composition of the treated water is also significant for the correct exploitation of high-manganese ores, because of the fact that non-oxidized iron can be competitively removed in relation to manganese. According to engineering practice it is considered as a technological problem when the surface of the catalytic bed in the manganese remover is covered with iron hydroxides, deactivating the catalytic activity of manganese ores media. The quartz bed for excessive iron removal is placed in the water treatment systems in line before the catalytic bed, so the manganese cycle can be extended several times [157]. The results of studies by Schaefer et al. [206] oppose that, proving pyrolusite oxidizes divalent iron ions to a trivalent form in an abiotic reaction, and the iron oxides coatings formed on its surface change its morphology, yet do not reduce redox activity. The authors emphasize that the results obtained do not coincide with the widespread belief that the pyrolusite catalytic effect is reduced by coatings of iron compounds and suggest that other compounds present in groundwater in a reduced form may be responsible for that. 

A critical operating parameter of the catalytic beds is backwashing with air and water, allowing the catalytic properties of the covered by deposits of precipitates from water surface to be renewed. As shown in Table 7 commercial manganese ores are characterized by various grain sizes (fine and thick) with a narrow or wide range of fractions. The high content of manganese dioxide shows that the materials are distinguished by high density and bulk density. Therefore, more water is used for backwashing than for other mineral media used for water treatment. Figure 1 evidently shows that twice the flow velocity is required to start the expansion of pyrolusite filter media than for commonly used quartz sand. This is a difficulty associated with the exploitation of filters filled with manganese ore, since too low intensity of backwashing leads to clogging of the filter bed and decrease of effectiveness of manganese removal from water.

## 5. Conclusions

Groundwater is widely used as drinking water worldwide, and its treatment is usually based on complex filter systems. This paper presented an overview of mineral materials applied in groundwater treatment relating to chemically and naturally coated filter media as well as to materials prepared in laboratory conditions and commercial filter media and full-scale filter systems. Commonly used quartz sand and chalcedonite are neutral filter media while anthracite and diatomite have some adsorption properties because of which they are not employed for groundwater treatment; similarly zeolite and glauconite are not used even though these materials are characterized by significant ion exchange capabilities. Groundwater treatment bases on the oxidation of water components and their retention in filter bed, in with the most important are oxyhydroxides of manganese and partly also iron. None of the above materials contain these ingredients, with an exception for the manganese ores. The others filter media gain redox activity as a result of coating in abiotic chemical or biotic/abiotic natural processes. The surface morphology also changes by the development of the specific surface area (very significant for low porosity materials) though research reports are not unequivocal in this matter. The appearance of MnO_x_ increases the ability to remove metals from water, however, this is mostly pronounced in the case of non-reactive inert materials. Already for zeolites with ion exchange capabilities, research data are ambiguous and sometimes the authors indicate that the removal possibilities are smaller after coating.

It is challenging to confirm the influence of MnO_x_ type on water treatment efficiency as a result of the very high impact in the systems of other factors like chemistry of treated water, biotic and/or abiotic nature of reaction in filter, aeration intensity, flow velocity and contact time, filtration bed geometry, and backwashing regime. However, independent research shows that high reactivity birnessite type manganese oxides, which is characterized by redox and cation exchange properties, play a crucial role in groundwater treatment by naturally coating filter media as well as the chemically formulated. Because of technological complexity, it is very demanding to determine the effect of the type of mineral material on the operation of a naturally coated filter media. The chemistry and specific surface area of the filter material can be significant for accelerating the start-up stage, although compared to other technological parameters it may be a rather less remarkable factor. Regardless of the oxide type and given the abiotic nature of the process, there is a certain relationship between its amount and exhaustion of manganese removal capacity. Filtration media containing several percent of MnO_x_ lose their ability to remove manganese (regeneration with an oxidizer) yet several dozen percentages of MnO_x_ give a possibility of continuous treatment in aerobic conditions with backwashing step. 

As emphasized in opinions, groundwater treatment based on aeration and filtration on naturally coating media and on manganese ores is recommended because of the exclusion of chemical additives and prevents the formation of oxidation by-products. All of these MnO_x_ filter media are able to remove manganese and iron from water based on various mechanisms. Manganese ore filter media are also capable of removing hydrogen sulphide and commercial chemically coated media additionally remove hydrogen sulphide, arsenic, and radium. Purification of water from manganese, iron, ammonium ion, arsenic, phosphorus, and antimony is possible using naturally coated filter media. However, the review shows that MnO_x_-coated media obtained in laboratory conditions are highly effective in removing heavy metals and metalloids from water. Considering the above, the question may be asked to what extent technologies based on MnO_x_ filter media can provide effective removal of heavy metals in operational conditions? This may be particularly important for groundwater treatment in degraded areas.

The chemistry, morphology, and structure of both the MnO_x_ coating and the mineral carrier affect the results of water purification. Water composition is another equally important variable. The multitude of factors makes it difficult to compare the suitability of filter media, especially when the experiments are designed variously. Most often the chemically coated filter media are evaluated in batch tests, while tests of under flow conditions are used rather for commercial coated filter media and manganese ores filter media. In case of naturally coated filter media, flow conditions are obvious and additionally the groundwater chemistry and microbiology are the major factors. The lack of defined standard test conditions makes it arduous to compare the suitability of materials for removing impurities from water, especially at the initial stage of laboratory batch tests. European Standards describe the methodologies for determining the manganese removal capacities of manganese ore (pyrolusite) and MnO_x_-coated glauconite (Manganese Greensand). There are also research works focused on standardizing the conditions for measuring the adsorption capacity of MnO_x_-coated media ripened in filters at water treatment plant. These methodologies are based on flow tests and the oxidation capacity concerns removal from water only the manganese. It is not comprehensive enough for other impurities. The methodologies also define the methods of preparing the water solution for testing, that is principal for investigation of groundwater treatment systems. This could be extended to procedures for the preparation of specific types of water.

## Figures and Tables

**Figure 1 materials-13-02232-f001:**
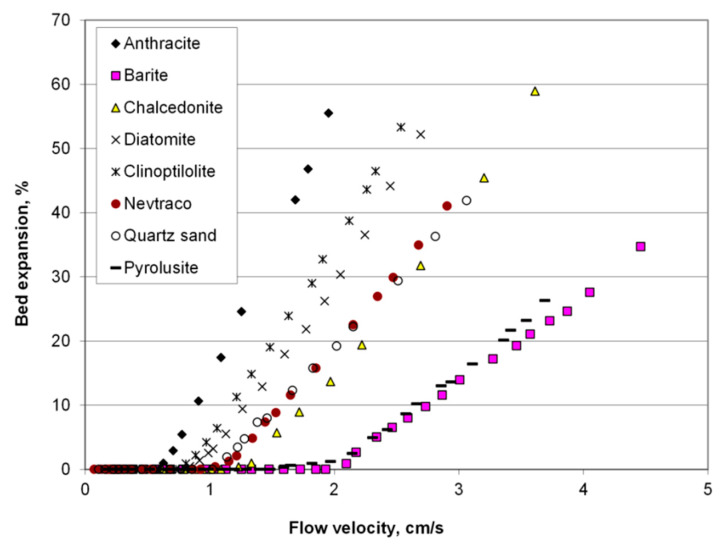
Expansion of mineral media in function of backwashing velocity, granulation of media 1.0–1.25 mm. Reprinted with permission from Ref. [71].

**Figure 2 materials-13-02232-f002:**
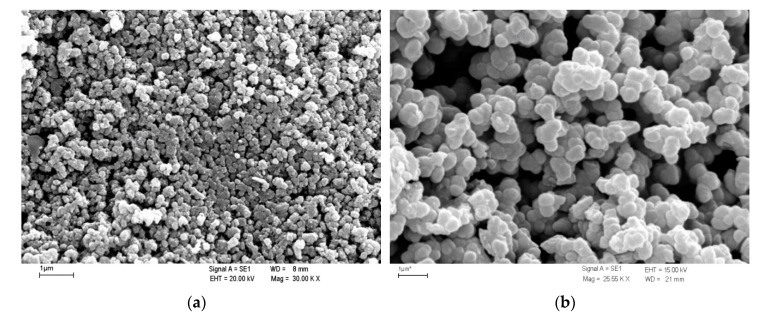
The newborn MnO_x_ clusters on the chalcedonite surface: (**a**) thermally bonded [120] and (**b**) chemically bonded [130]. Reprinted with permission from Ref. [120,130].

**Figure 3 materials-13-02232-f003:**
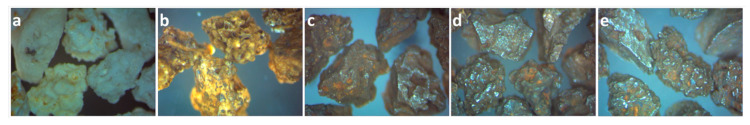
The changes of surface presence between raw (**a**) and once (**b**), twice (**c**), three times (**d**), and four times (**e**) coated chalcedonite (magnification 40×). Reprinted with permission from Ref. [133].

**Figure 4 materials-13-02232-f004:**
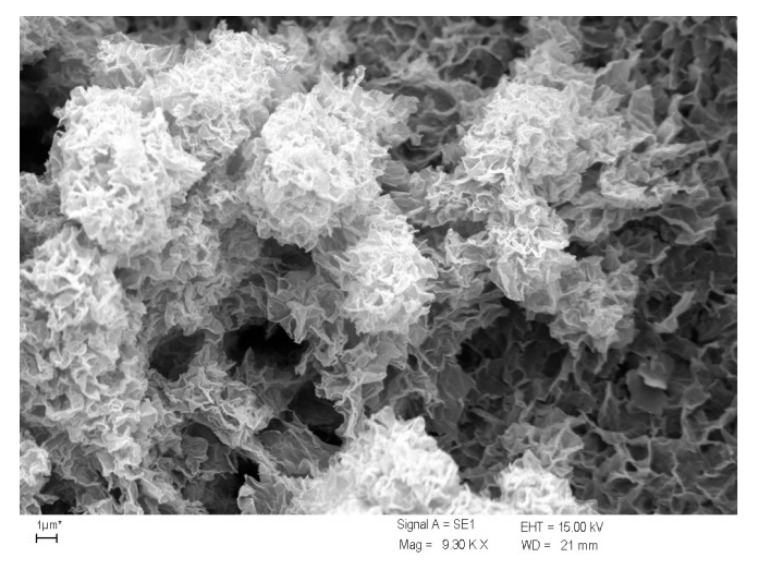
The surface of MnO_x_ coated chalcedonite after removal manganese from water. Reprinted with permission from Ref. [130].

**Table 1 materials-13-02232-t001:** Properties of mineral materials using as filter media for water treatment.

Material	Density (kg/m^3^)	Bulk Density (kg/m^3^)	Specific Surface Area (m^2^/g)	Mohs Hardness (-)
Quartz sand	2650 [44]	1520–1568 [45]	0.05–1.8 [46]	7 [47]
	2650–2667 [45]			
Chalcedonite	2390–2500 [48]	1237–1403 [45]	6.13 [49], 7.44 [50]	6–6.5 [51]
	2488–2682 [45]			
Diatomite	2000–2100 * [52]	200–400 * [52]	14.6 [53],	5–6.5 [54]
	2244 ** [55]	1280–1780 ** [56]	22 [57], 42 [58]	
Glauconite	2450–2600 [59]	1380 [60]	48 [58], 58 [61], 78 [60]	2 [47]
	2650–2750 [62]			
Zeolite	2339–2407 (clinoptilolite) [45] 2390 (clinoptilolite) [63]	800–1000 (clinoptilolite) 700–850 (chabazite) 1000–1100 (phillipsite-analcime) [64]	14 (clinoptilolite 71.8 wt. %) [57] 19.2 (clinoptilolite 45–50 wt. %) [65] 30.98 (clinoptilolite >90 wt. % ) [66] 60.32 (phillipsite and chabazite 55 wt.% ) [67]	3.5–4 (clinoptilolite) 4–5 (chabazite) 4–4.5 (phillipsite) 5–5.5 (analcime) [47]
Anthracite	1400–1450 [68,69] 1743–1801 [45]	730 [69] 732–900 [45]	2.2, 6.4 [70]	2–4 [54]

***** pure, fine-particle sized, ****** compact, low-porous rock.

**Table 2 materials-13-02232-t002:** Characteristics of chemically coated mineral media.

Carrier	Coating Components	Coating Method *	Specific Surface Area before/after Coating of	Amount of Coating on the Carrier	Maximum Adsorption Capacity of Coated Material	Ref.
Quartz sand	Pyrolusite (MnO_2_), γ-Mn_2_O_3_, Mn(OH)_4_	Reduction method: prepared under moderate temperature and acidic conditions by impregnation processes with igneous rock and KMnO_4_	–/1.99 m^2^/g	0.396 mg Mn/g	0.3805 mg Mn/g batch conditions, 25 °C	[121]
Quartz sand	Amorphous manganese dioxide	Reduction method: acid-digested sand placed in KMnO_4_ boiling solution, MnO_x_ precipitated by dropwise 37.5% HCl addition, 1 h contact time with agitation	0.674/0.712 m^2^/g	5.46 mg Mn/g	0.00591 mmol Cu/g (0.375 mg Cu/g) 0.00771 mmol Pb/g (1.60 mg Pb/g)batch conditions, 15 °C and 22 °C resp.	[122,123]
Chalcedonite	Amorphous manganese dioxide	Redox method: chalcedonite soaked in 20% MnSO_4_, next soaked in 5% KMnO_4_ to MnO_x_ precipitate	6.13/9.88 m^2^/g	–	1.07 mg Mn/g 10.3 mg Pb/g batch conditions, 10 °C 0.76 g Mn/L (0.62 mg Mn/g) ** flow conditions	[49,115,124,125]
Chalcedonite	Birnessite type manganese oxide	Oxidation method: chalcedonite soaked in 6 M NaOH (2 h, T 80–90 °C), next soaked in 2.5 M MnCl_2_ (pH 1–2, 10 h, room T), soaked in 6 M NaOH (10 h, room T) and finally oxidized in air	6.13/10.25 m^2^/g	–	2.63 g/L (2.16 mg Mn/g) ** flow conditions	[49,114]
Diatomite	Birnessite type manganese oxide	Oxidation method: chalcedonite soaked in 6 M NaOH (2 h, T 80–90 °C), next soaked in 2.5 M MnCl_2_ (pH 1–2, 10 h, room T), soaked in 6 M NaOH (10 h, room T) and finally oxidized in air	33/80 m^2^/g	0.38 g MnO_x_/g	99 mg Pb/g batch conditions, 23 ± 1 °C	[126]
Diatomite	Amorphous manganese dioxide	Reduction method: carbon covered diatomaceous earth dispersed into KMnO_4_ solution, next microwave heated (10 min, 700 W) and cooled	23.3/24.1 m^2^/g	–	56.843 mg Pb/g batch conditions, 30 °C	[127]
Zeolite: clinoptilolite 36%, mordenite 33%, quartz 26%, montmorillonite 5%	Vernadite (δ-MnO_2_)	Reduction method: Na-zeolite placed in KMnO_4_ solution (T 90 °C), manganese oxide precipitated by dropwise 37.5% HCl addition, contact time 1 h with agitation	–	–	1.123 meq Mn/g (30.85 mg Mn/g) batch conditions, 25 °C	[41]
Zeolite: 58–75% of clinoptilolite	Cryptomelane (α-MnO_2_)	Redox method: Mn^2+^-clinoptilolite treated with KMnO_4_ solution (T 20 °C)	–	MnO_2_ 0.30–0.49% (1.9–3.1 mg Mn/g)	6.9–21.6 mg Mn/g batch conditions, 20 ± 1 °C	[119]

In parentheses are placed the converted values; * each procedure was completed with a stage of washing by deionized water and drying; ** calculated from g Mn/L of bed to mg Mn/g of bed using bulk density of chalcedonite equals 1219 kg/m^3^ [128].

**Table 3 materials-13-02232-t003:** Experimental procedures for comparing manganese removal ability of MnO_x_ filter media.

Specified Parameter with Unit	Adsorption Capacity or Uptake Capacity (mg Mn/mg of Media)	Oxidation Capacity (mg Mn/L of Media)	Oxidation Capacity (L of Treated Water/L of Media)
Test type	column	column	column
Type of media	MnO_x_-coated media exploited in filters enhanced with chemical oxidants	MnO_x_-coated glauconite (Manganese Greensand)	manganese ore (pyrolusite)
Flow velocity (m/h)	24.5	10	10
Concentration of manganese in test water (mg Mn/L)	0.3–0.5	10	1
Composition of test water	pH 6.3 ± 0.1 MnSO_4_, NaHCO_3_ and CaCl_2_ (alkalinity 25 mg CaCO_3_/L, calcium 10 mg/L)	pH 6–7 MnSO_4_ (27.5 mg/L), NaHCO_3_ (100 mg/L)	pH 7.0 ± 0.5 MnSO_4_ (2.75 mg/L), NaHCO_3_ (20 mg/L)
Amount of test sample	2–10 g	300 mL	20 g
Additional sample treatment	sieving, washing, regeneration with chlorine solution (20 mg/L, pH 6.3)	washing, regeneration with KMnO_4_ (3.0 g/L)	sieving, washing, regeneration with chlorine solution (1200 mg/L)
References	[143]	[93]	[142]

**Table 4 materials-13-02232-t004:** Features of commercial MnO_x_ coated filter media. Adapted from [128].

Features	Trade Name			
	Manganese Greensand	GreensandPlus^TM^	MTM^®^	BIRM^®^
Carrier compounds (wt.%)	glauconite 96–97	quartz 90.4–93.6	silicon dioxide >75	quartz 40–60
			Quartz <10	
			cristobalite <0.1	
Coating compounds (wt.%)	manganese oxide 3–4	manganese dioxide 3.2–4.8	manganese dioxide <1	manganese dioxide 10–20
Regenerative agent	potassium permanganate	chlorine	potassium permanganate or chlorine	does not require
Dissolved components possible to remove from water	Iron Manganese hydrogen sulphide	Iron Manganese hydrogen sulphide arsenic, radium	Iron Manganese hydrogen sulphide	Iron manganese
Removal capacities (g/L of media)	Fe: 1.34	Fe: 1.34	Fe: 1.34	not specified
	Fe+Mn: 0.94	Mn: 0.67	Mn: 0.67	
	H_2_S: 0.40	H_2_S: 0.27	H_2_S: 0.27	
Max. concentration of components in water (mg/L)	Fe: 15	not specified	Fe: 15	H_2_S absence
	Mn: 15		Mn: 5	TOC 4–5
	H_2_S: 2		H_2_S: 2	Cl_2_ 0.5
Preferred water pH	6.2–8.5	6.2–8.8	6.2–8.5	6.8–9.0
Density (kg/m^3^)	2400–2900	2400	2000	2000
Bulk density (kg/m^3^)	1382	1410	720–800	580–610
Effective size *d*_10_ (mm)	0.3–0.35	0.3–0.35	0.43	0.48
Uniformity coefficient	1.6	1.6	2.0	2.7
Flow velocity (m/h)	9–15	5–12	6–15	10.5–15
Min. bed depth (m): single layer double layer	0.76	0.76	0.90	0.90
	–	0.40–0.45	0.60	0.75
Manufacturer	Inversand Co.	Inversand Co.	Clack Corp.	Clack Corp.
References	[147,148]	[149,150]	[151,152]	[153,154]

**Table 5 materials-13-02232-t005:** Conditions of the start-up during naturally coating of the filter media. Adapted from [128].

Time of Start-up	Type of Media and Grain Size	Technological Conditions	Chemistry of Raw Water	Ref.
20 days	Quartz sand and gravel 2–5 mm	Empty bed contact time: 10 min bed depth: 0.2 m backwashing with water inoculation with manganese oxidized bacteria	pH 7.0–7.2	[165]
			Mn: 1.5–2 mg/L	
			Fe: 5–6 mg/L	
			O_2_: 3.5 mg/L	
20 days	Chalcedonite 0.5–2.0 mm	Flow velocity: 5.5 m/h empty bed contact time: 8 min bed depth: 0.75 m backwashing with treated water	pH 7.2 ± 0.03	[128]
			Eh +210 ± 29 mV	
			Mn: 0.241 ± 0.024 mg/L	
			Fe: 0.03 ± 0.01 mg/L	
			NH_4_^+^: 0.07 ± 0.04 mg/L	
			O_2_: 3.39 ± 0.73 mg/L	
21 days	Quartz sand 0.5–1.8 mm	Flow velocity: 5.5 m/h empty bed contact time: 8 min bed depth: 0.75 m backwashing with treated water	pH 7.2 ± 0.03	[128]
			Eh +210 ± 29 mV	
			Mn: 0.241 ± 0.024 mg/L	
			Fe: 0.03 ± 0.01 mg/L	
			NH_4_^+^: 0.07 ± 0.04 mg/L	
			O_2_: 3.39 ± 0.73 mg/L	
25 days	Quartz sand 0.7–1.25 mm	Flow velocity: 5.1 m/h empty bed contact time: 3.5 min bed depth: 0.3 m backwashing with water	pH 7.5–7.9	[176]
			Eh +200–+290	
			Mn: 0.10–0.15 mg/L	
			Fe: 0.03–0.1 mg/L	
			NH_4_^+^: max 0.2 mg/L	
			O_2_: 8.0–9.5 mg/L	
26 days	Quartz sand 1.0 mm	Flow velocity: 7.0 m/h empty bed contact time: 8.6 min bed depth: 1.0 m backwashing with water and air KMnO_4_ dosage	pH 8.0 ± 0.1	[178]
			Mn: 0.99 ± 0.12 mg/L	
			Fe: 1.06 ± 0.2 mg/L	
			NH_4_^+^: 1.39 ± 0.1 mg/L	
			O_2_: 6.5–7.0 mg/L	
40 days	Chalcedonite d_10_ 0.8–1.0 mm UC 1.4–1.6	Flow velocity: 6–12 m/h bed depth: 1.8 m	pH 6.9–7.5	[186]
			Mn: 0.1–0.9 mg/L	
			Fe: 0.4–5.0 mg/L	
			NH_4_^+^: 0.2–0.9 mg/L	
			O_2_: 5.5–11.0 mg/L	
70–80 days	Chalcedonite 0.8–2.4 mm	Flow velocity: 11–14 m/h bed depth: 0.8 m backwashing with water and air	Mn: 0.22–0.27 mg/L	[190]
			NH_4_^+^: 0.7–0.8 mg/L	
7 months	Sand 0.5–1.2 mm	Flow velocity: 1.5 m/h empty bed contact time: 40 min bed depth: 1.00 m inoculation with indigenous bacteria backwashing with treated, chlorinated water	pH 6.7–6.9	[164]
			Mn: 0.8–2.0 mg/L	
			Fe: 0.15–0.20 mg/L	
			O_2_: present	
350 days	Anthracite and quartz sand	Flow velocity: 6–8 m/h bed depth: 1.0 m anthracite and 0.6 m quartz sand backwashing with air and treated, chlorinated water	pH 6.9–7.3	[191]
			Mn: 0.21–0.30 mg/L	
			Fe: 0.55–1.63 mg/L	
			NH_4_^+^: 0.65–0.90 mg/L	

**Table 6 materials-13-02232-t006:** The elemental composition of the surfaces of naturally coated filter media.

Location of Water Treatment Plant	De Punt, The Netherlands	Onnen, The Netherlands	Wierden, The Netherlands	Poznań, Poland	Poznań, Poland	Joyo, Japan	Joyo, Japan
**Operating time of filter**	15 years	40 years	18 years	10 years	10 years	3 years	15 years
**Place of origin**	from 1.4–1.5 m depth of manganese removal zone	from the top of manganese removal postfilter	from the top of manganese removal postfilter	from filter iron removal zone	from filter manganese removal zone	from the top of iron and manganese removal filter	from the top of iron and manganese removal filter
**Carrier**	quartz sand	quartz sand	quartz sand	quartz sand	quartz sand	anthracite	anthracite
**Content of major elements, wt.%**							
**Fe**	n.a.	n.a.	n.a.	29.5	15.6	9.21	23.13
**Mn**	n.a.	n.a.	n.a.	18.5	21.5	12.87	11.58
**O**	n.a.	n.a.	n.a.	46.0	55.0	29.72	26.56
**C**	n.a.	n.a.	n.a.	n.a.	n.a.	36.06	24.84
**Content of side elements, wt.%**							
**Ca**	2.3	7.2	7.7	3.00	2.90	10.06	9.71
**Si**	5.9	2.6	0.9	1.05	0.90	0.69	1.87
**Al**	<0.1	0.6	<0.1	0.70	3.51	0.19	n.d.
**Mg**	<0.1	0.4	0.3	n.a.	n.a.	n.a.	n.a.
**Na**	<0.1	0.2	<0.1	n.a.	n.a.	n.a.	n.a.
**K**	<0.1	<0.1	<0.1	n.d.	n.d.	0.15	n.d.
**S**	n.a.	n.a.	n.a.	0.09	n.d.	0.50	0.66
***p***	n.a.	n.a.	n.a.	0.40	0.21	0.56	1.63
**Analysis technique**	-	-	-	SEM/EDX	SEM/EDX	WDXS	WDXS
**Ref.**	[183]	[183]	[183]	[194]	[194]	[167]	[167]

n.a.—Not analyzed, n.d.—Not detected.

**Table 7 materials-13-02232-t007:** Features of commercial catalytic filter media consist of manganese ore. Adapted from [128].

Features	Trade Name					
	G1	Defeman	Multiman 3M	Pyrolox	Filox-R	MetalEase
MnO_2_ content (wt.%)	≥82	84	min. 80	–	75–85	75–85
Preferred water pH	≥7	7.0–8.5	>7.4	6.5–9.0	6.5–9.0	5.0–9.0
Max. concentration of components in water (mg/L)	for Fe, Mn, (values not given)	Fe: 20 Mn: 1.5	Fe: 15 Mn: 1.5	for Fe, Mn, H_2_S (values not given)	Fe: 10 Mn: 5 H_2_S: 3	Fe: 10 Mn: 3 H_2_S: 5
Flow velocity (m/h)	10–20	to 20	7–15	12	15	10–12
Min. depth of bed (cm)	35–45	–	–	46	51	61
Typical grain size (mm)	1–3	0.5–0.8 0.8–3.0 3.0–10.0	0.8–2.5 1.0–3.0	0.42–0.84 0.84–2.38	0.42–1.68	0.42–1.68
Density Bulk density (kg/m^3^)	4100–4300	–	4000–4200	3800	–	–
	1800–2000	1900	2000	1920	1760	1824
Manufacturer	Global Concepts 2000 Polska Sp. z o.o.	Global Concepts 2000 Polska Sp. z o.o.	Dynamik Filtr s. j.	Prince Minerals Inc.	Watts Water Technologies EMEA B.V.	Safe Water Technologies Inc.
References	[203]	[207]	[208]	[209]	[210]	[205]

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
