# Peer review of "Mineral Materials Coated with and Consisting of MnOx—Characteristics and Application of Filter Media for Groundwater Treatment: A Review"

_materials, 2020, doi:10.3390/ma13102232_

Round 1

Reviewer 1 Report

The paper is on the groundwater treatment which is undoubtedly is a relevant issue. The review provides the summarized characterization of well-known natural and synthetic materials used for the water treatment, covering those employed in full-scale filter systems. The differences in the efficiency and treatment mechanism are shown for a variety of sorbents. For the commercially applied materials, the authors provide the references to commercial date sheets and emphasize the application of these materials in Poland and local documents. The discussed issues include the techniques of depositing manganese oxides on the carriers; the factors influencing the quality of coating and sorption capacity of the filter materials; commercial MnOx-containing products, etc. In general, the review summarizes recent investigations on the groundwater treatment using MnOx-containing filter materials. Most of the papers cited are relatively recent (published within the last 5 years). This study is well-made and may be interesting for the researchers working on the development of novel filter materials and related areas. I recommend this review article for publication in its present form.

Author Response

Dear Reviewer,

Thank you very much for your engagement in evaluation the submitted manuscript. We are very pleasant that you appreciated our work and you think it is important.

Best Regards,

Authors

Reviewer 2 Report

This manuscript by Michel et al. aims to review the use of minerals coated with manganese oxides or manganese ores for the treatment of groundwater. This manuscript is a nice piece of work and it scope fits well with the readership of Materials. However, in my view, this manuscript suffers from several limitations that should be corrected before publication; (i) the language of the manuscript is in some parts very difficult to understand and I suggest an in-depth proofread of the manuscript, (ii) the structure of the manuscript is sometimes hard to follow with some sections that are very long and subsections would be beneficial, (iii) the groundwater treatment is not in the heart of this work, but more precise information of the removal of contamination as a function of the used materials (even if it depends on the level of contamination of groundwater itself) would be beneficial, (iv) the conclusions needs to be improved as it is currently too methodological, without any precise recommendation.

Based on these comments, I recommend major modifications on this manuscript.

Specific comments are given below.

The title of the manuscript should be modified as it does not reflect the content of the manuscript, please use a more general title as your review also focuses on manganese ores and mineral filters.

The abstract is quite short and is not properly written. The second sentence (L17-20) should be clarified, and the main objectives of your work as well. As well, the last sentence, L25-26 is very weird and should be corrected.

L31-35, here you only cite references on the inorganic ions removal by minerals, (if we except petroleum substances) however, plenty of reviews have been made on the removal of organic compounds, please indicate at least one of them.

L48 Here you seem to identify some targets “iron, manganese, ammonium, hydrogen sulphide” however this is not clearly expressed throughout the manuscript (as for example in the conclusions). What is the precise added values of manganese oxide coated materials for the groundwater treatment?

L49-51 this sentence is unclear and should be rewritten

L51-53 this sentence is also not correct in its grammar

L74 this sentence does not make any sense, please correct

Section 2: This section is too long (6 pages) and verbose with plenty of details that are not at the heart of your work (systematically the Mohs scale for example). I suggest to gather the most important data (SSA, CEC, density, etc. in a table allowing a direct comparison rather than long discussion that are hard to compare.

About Chalcedonite and Diatomite, for the description of these two rocks, you extensively cite references in polish language. It would be ok, but here you cite almost exclusively articles in this language. Also you constantly refer to mid-european deposits, even if diatomite and chalcedonite are worldwide present rocks. I suggest to widen the description of these two rocks, out of your country. Also, the term spongliolite does not exist, please modify.

L236-237 why distinct “lead, cadmium copper and copper” and radioisotopes? Both are cations, and in view of the cation exchange capacity of glauconite, it is obvious that there would be an affinity for cations, please simplify the writing.

L237-239 Several times in the manuscript, the comparison/presentation are not at the same scale. Here you give one value for the cation exchange capacity of glauconite (17meq/100g) even this is variable, and then you present successively three close values of SSA please be more consistent for the presentation of the data (I suggest to present only min-max SSA for brevity).

L245 What does mean “is for example 1380 kg/m3”?

L268-270 “The characteristic structural property of zeolites, making them molecular sieves, is not used in the treatment of groundwater.” I don’t understand this sentence? So zeolite are not used for the treatment of groundwater? If yes, how can we hinder a structural property of the material?

L273 “mg Mn/L” This is a curious unit for an adsorption capacity? Can you explain?

L279 “then” rather of than

L296-297 here you indicate that the minimal CEC should be 1,2mgNH3/100g although earlier (see above comment) you claim that CEC is not used in groundwater? Could you explain?

L323 “post-coagulant” you use some terms that were not previously contextualized. It’s okay if you consider that all the readers are specialized on water treatment but it is not the case!

Section 3.1.1 covers 4 pages of text with 2 tables and 3 figures without any subsection, it is very hard to follow. For example, cut the section for various synthesis conditions (experimental, operating). This is mandatory for clarity.

L364 “five times greater” in general in the manuscript you did not present any kinetic data, however, compare adsorption capacities sole is not consistent for the filtration purpose.

L372-373 “values determined under flow conditions” please reword

I’m not particularly convinced by the distinction between section 3.1.1 and 3.1.2, as commercial products are coated materials too? Please explain?

L543 “two out of five” there is four commercial materials? Where is the fifth?

L545-547 This sentence is very weird and needs to be better connected in this section.

L579 what is the “manganese effect”?

L581-582 I believed that larger hydraulic resistances would lengthen the filtration time?

L605-626 I don’t understand why is this section here. You compare the efficiency of commercial and non-commercial filters. It would be presented in a separated section.

L628 and after, in this section you mention for the first time biological processes, yet it is not specific to naturally coated filter media and such phenomenon can occur with the other materials. I suggest to mention such phenomenon in the other sections.

L632-633 “which participate in simultaneous transformation of iron and manganese [138,143,144] as well as iron, manganese and ammonium ions” please reword as it does not make any sense

L644-647 This conclusion/assumption is interesting but is not enoughly discussed in relation with the literature. Here you write X concludes … Y writes the opposite … Z writes another thing so I conclude… please provides understandable information to support your interesting conclusion.

L648-649 “popular” “frequently used” in which context? Scientific literature? Field operation? Please be more specific

L650 I’m not sure that this section is the most appropriate to introduce a new material “melaphyre” please only cite those previously introduced.

L652 “iron and manganese oxyhydroxides” in this section it may be highlighted that the natural coating is iron+manganese rather than only manganese. It is therefore hard to compare with lab-MnOx coated materials.

L655 “probably on properties” please reword this sentence

L658-659 this sentence is too vague even it introduces an interesting assumption, please be clearer.

L674 “50-90%” really 90%?

L682-683 Chemically coated filter media does not present the potential for natural coating when used in field operations? Can you assume that chemically coated filter media used for groundwater treatment could be naturally coated by MnOx and FeOx? Please explain?

L708 “iron and manganese removal zones” you use this expression several times throughout the manuscript, however, you never clearly explain this notion. As the readers will not be only specialists, a section explaining this may be useful (see comment above on other process)

Table 5 Could be present Fe and Mn % in coatings? It would be more interesting than Na content?

Section 4, same comments on the structure and subsection (commercial/ores, experimental/operating) than for section 3

Your conclusions are interesting but may be improved. I have the feeling that you are stick on very technical questions, and you did not conclude on the best materials for the groundwater treatment (even if I agree that such conclusion can not be such definitive).

Author Response

Dear Reviewer,

Thank you very much for your engagement in evaluation the submitted manuscript. We appreciate your accurate text analysing. Your work helped us to avoid making mistakes and misunderstandings. We hope that our article is better in its actual form. We put detailed answers below. In second version of the manuscript the corrections to your comments are marked by red. We are aware of language failures so a correction will be made by the appropriate editor. The following answers are also included as an attachment.

Best Regards,

Authors

C-Reviver’s comment

A-author’s answer

L-line number in corrected text

C1. The title of the manuscript should be modified as it does not reflect the content of the manuscript, please use a more general title as your review also focuses on manganese ores and mineral filters.

A1. We hope the new title is more general:

L 2-4: “Mineral Materials Coated with and Consisting of MnOx – Characteristics and Application of Filter Media for Groundwater Treatment: A Review”

C2. The abstract is quite short and is not properly written. The second sentence (L17-20) should be clarified, and the main objectives of your work as well. As well, the last sentence, L25-26 is very weird and should be corrected.

A2. The abstract is rewritten. Its volume is close to maximal 196 words/200words:

L 18-32: “For groundwater treatment the technologies involving oxidation on MnOx filter bed are beneficial, common and effectively used. The presence of MnOx is the mutual feature of filter media, both: MnOx coated mineral materials like quartz sand and gravel, chalcedonite, diatomite, glauconite, zeolite or anthracite as well as consisting of MnOx manganese ores This review bases on analysis of research and review papers, commercial data sheets and standards. The aim of the paper was to provide new suggestions and useful information for further investigation of MnOx filter media for groundwater treatment. The presented compilations are based on characteristic of coatings, methods and conditions of its obtaining and type of filter media. The relationship between the properties of MnOx amendments and the obtained purification effects as well as the commonly used commercial products, their features and applications, have been discussed. It was concluded about improving catalytic/adsorption properties of non-reactive siliceous media opposed to ion-exchange minerals and about possible significance of birnessite type manganese oxide for water treatment. Research needs related to the assessment of the use MnOx filter media to heavy metals removal from groundwater in field operations were identified and also to standardize methodology of testing MnOx filter media for water treatment.”

C3. L31-35, here you only cite references on the inorganic ions removal by minerals, (if we except petroleum substances) however, plenty of reviews have been made on the removal of organic compounds, please indicate at least one of them.

A3. We added two review articles generalized information about dyes and organic micro-pollutants removal on clay minerals and zeolites respectively. The third article takes about hazardous oxyanions, the forms of inorganic pollutants hard to removal. We added two review articles generalized information about dyes and organic micro-pollutants removal on clay minerals and zeolites respectively. The third article takes about hazardous oxyanions, the forms of inorganic pollutants hard to removal. This work is also a review.

L 38: Adeyemo, A.A.; Adeoye, I.O.; Bello, O.S. Adsorption of dyes using different types of clay: a review. Appl. Water Sci. 2017, 7, 543–568, doi:10.1007/s13201-015-0322-y.

Jiang, N.; Shang, R.; Heijman, S.G.J.; Rietveld, L.C. High-silica zeolites for adsorption of organic micro-pollutants in water treatment: A review. Water Res. 2018, 144, 145–161, doi:10.1016/j.watres.2018.07.017.

Weidner, E.; Ciesielczyk, F. Removal of Hazardous Oxyanions from the Environment Using Metal-Oxide-Based Materials. Materials 2019, 12, 927, doi:10.3390/ma12060927.

C4. L48 Here you seem to identify some targets “iron, manganese, ammonium, hydrogen sulphide” however this is not clearly expressed throughout the manuscript (as for example in the conclusions). What is the precise added values of manganese oxide coated materials for the groundwater treatment?

A4. We determined that iron, manganese, ammonium ion and hydrogen sulphide are the target components for technologies using described materials:

L 59-61: “Iron, manganese, ammonium ion and hydrogen sulphide are common components of groundwater and can be removed by classic technologies based on manganese oxides coated and consisted of filter media.”

C5. L49-51 this sentence is unclear and should be rewritten

A5. The sentence have been reworded:

L 54-55: “According to the World Health Organization guidelines for drinking water quality [14], its components can not constitute a threat to human health.”

C6. L51-53 this sentence is also not correct in its grammar

A6. The sentence have been reworded:

L 61-63: “The excessive concentration of these components leads to problems such as undesirable taste, odour and colour of water, staining laundry and clogging of fittings by sediment and biofilm formation [6].”

C7. L74 this sentence does not make any sense, please correct

A7. The sentence about scientific opinion was rejected. Generally the paragraph was reorganized.The sentence after re-editing is:

L 79-81: “The particular utility is due to the fact that under optimized conditions this kind of treatment does not lead to the formation of toxic residues which may arise in the use of chemical oxidants such as chlorine or ozone.”

C8. Section 2: This section is too long (6 pages) and verbose with plenty of details that are not at the heart of your work (systematically the Mohs scale for example). I suggest to gather the most important data (SSA, CEC, density, etc. in a table allowing a direct comparison rather than long discussion that are hard to compare.

A8. Chapter 2 has been redrafted. It is shorter over by one page, the data is summarized in the table. With blue colour we marked sentences which was changed after cutting excessive information and after transferring data to the table 1.

C9. About Chalcedonite and Diatomite, for the description of these two rocks, you extensively cite references in polish language. It would be ok, but here you cite almost exclusively articles in this language. Also you constantly refer to mid-european deposits, even if diatomite and chalcedonite are worldwide present rocks. I suggest to widen the description of these two rocks, out of your country. Also, the term spongliolite does not exist, please modify.

A9. The description is rewritten. We deleted description about Polish deposits. We hope now the description is more general. We added references:

Filtration materials for groundwater: a guide to good practice; Kozyatnyk, I., Ed.; IWA Publishing: London, 2016; ISBN 978-1-78040-699-2.

              Haldar, S.K. Introduction to mineralogy and petrology; Elsevier: Amsterdam, 2014; ISBN 978-0-12-408133-8.

              Benkacem, T.; Hamdi, B.; Chamayou, A.; Balard, H.; Calvet, R. Physicochemical characterization of a diatomaceous upon an acid treatment: a focus on surface properties by inverse gas chromatography. Powder Technol. 2016, 294, 498–507, doi:10.1016/j.powtec.2016.03.006.

              Flores-Cano, J.V.; Leyva-Ramos, R.; Padilla-Ortega, E.; Mendoza-Barron, J. Adsorption of Heavy Metals on Diatomite: Mechanism and Effect of Operating Variables. Adsorpt. Sci. Technol. 2013, 31, 275–291, doi:10.1260/0263-6174.31.2-3.275.

Zhao, Y.; Tian, G.; Duan, X.; Liang, X.; Meng, J.; Liang, J. Environmental Applications of Diatomite Minerals in Removing Heavy Metals from Water. Ind. Eng. Chem. Res. 2019, 58, 11638–11652, doi:10.1021/acs.iecr.9b01941.

Galal Mors, H.E. Diatomite: Its Characterization, Modifications and Applications. Asian J. Mater. Sci. 2010, 2, 121–136, doi:10.3923/ajmskr.2010.121.136.

              Ibrahim, S.S.; Selim, A.Q. Heat treatment of natural diatomite. Physicochem. Probl. Miner. Process. 48, 413–424.

We have problems with worldwide searching. References to chalcedonite are still of Polish authors, but in English and journals available on-line. Also we reject from references monography in Polish.

              Radziemska, M.; Koda, E.; Bilgin, A.; Vaverková, M. Concept of Aided Phytostabilization of Contaminated Soils in Postindustrial Areas. Int. J. Environ. Res. Public. Health 2017, 15, 24, doi:10.3390/ijerph15010024.

              Naziemiec, Z.; Pichniarczyk, P.; Saramak, D. Methods of improvement chalcedonite processing effectiveness with the use of density separation. Gospod. Surowcami Miner. 2017, 33, 163–178, doi:10.1515/gospo-2017-0038.

C10. L236-237 why distinct “lead, cadmium copper and copper” and radioisotopes? Both are cations, and in view of the cation exchange capacity of glauconite, it is obvious that there would be an affinity for cations, please simplify the writing.

A10. Chapter 2 has been corrected. Present form after re-editing:

L 227-230: “Glauconite can be chemically regenerated and in this form effectively removes metals from water, however, raw glauconite exhibits also exchange capacity. For this reason, glauconite can be used for water purification [57,61,90–92].”

C11. L237-239 Several times in the manuscript, the comparison/presentation are not at the same scale. Here you give one value for the cation exchange capacity of glauconite (17meq/100g) even this is variable, and then you present successively three close values of SSA please be more consistent for the presentation of the data (I suggest to present only min-max SSA for brevity).

A11. In text we tried to generalize. In table 1 are ranges when reference inform about range of values, and only values when reference inform just about particular value.

C12. L245 What does mean “is for example 1380 kg/m3”?

A12. We deleted “for example”, the value is in table 1

C13. L268-270 “The characteristic structural property of zeolites, making them molecular sieves, is not used in the treatment of groundwater.” I don’t understand this sentence? So zeolite are not used for the treatment of groundwater? If yes, how can we hinder a structural property of the material?

A13. We rejected the sentence: “The characteristic structural property of zeolites, making them molecular sieves, is not used in the treatment of groundwater.”

We wanted to say that in groundwater treatment in one filter are not applied different zeolites with different channel sizes to remove molecules of different sizes. But it was actually badly written and unnecessary.

C14. L273 “mg Mn/L” This is a curious unit for an adsorption capacity? Can you explain?

A14. Of course, this is a serious mistake. Thank you for finding. However, we deleted this data to shorten chapter 2.

C15. L279 “then” rather of than

A15. We deleted the sentence with this word to shorten chapter 2.

C16. L296-297 here you indicate that the minimal CEC should be 1,2mgNH3/100g although earlier (see above comment) you claim that CEC is not used in groundwater? Could you explain?

A16. Explained above in A13. The false sentence was thrown out.

C17. L323 “post-coagulant” you use some terms that were not previously contextualized. It’s okay if you consider that all the readers are specialized on water treatment but it is not the case!

A17. The information about genesis of the suspension in not important in the general context. We deleted the sentence to shorten chapter 2.

C18. Section 3.1.1 covers 4 pages of text with 2 tables and 3 figures without any subsection, it is very hard to follow. For example, cut the section for various synthesis conditions (experimental, operating). This is mandatory for clarity.

A18. We changed the subsections as follows:

  1. Manganese oxides coated filter media for groundwater treatment

              3.1. Chemically coated filter media

                            3.1.1. Coatings characteristic

                            3.1.2. Adsorption properties

                            3.1.3. Commercial media characteristic

                            3.1.4. Operating conditions and examples

              3.2. Naturally coated filter media

                            3.2.1. Coating formation

                            3.2.2. Coatings characteristic

  1. Manganese oxides consisted of filter media for groundwater treatment

              4.1. Manganese ores properties

              4.2. Commercial media characteristic

              4.3. Operating conditions

C19. L364 “five times greater” in general in the manuscript you did not present any kinetic data, however, compare adsorption capacities sole is not consistent for the filtration purpose.

A19. We added general information about kinetics of adsorption on MnOx coated media:

L 392-395: “Kinetics studies of heavy metals and metalloids adsorption on diverse MnOx coated mineral media (quartz sand, chalcedonite, diatomite, zeolite) confirm that process follows a pseudo-second-order model and is chemisorption [41,122,125,134–136]. In analyzed systems adsorption was endothermic in nature and occurred spontaneously.”

C20. L372-373 “values determined under flow conditions” please reword

A20. Reworded sentence:

L 320-323: “While carrying out research in the column tests it has been demonstrated that chalcedonite coated by birnessite type manganese oxide has more than three times greater adsorption capacity [114] than coated with amorphous manganese dioxide [115] (Table 2).”

C21. I’m not particularly convinced by the distinction between section 3.1.1 and 3.1.2, as commercial products are coated materials too? Please explain?

A21. Yes, you have a right, the commercial products are coated materials. Sections has been changed – presented in answer 18.

C22. L543 “two out of five” there is four commercial materials? Where is the fifth?

A22. This sentence applies to the work of other authors (they analyzed 5 materials). Does not apply to the table data - there are actually 4 materials. We have corrected the text and hope it is more clear now.

L 498-505: “Commercial materials, distributed worldwide as filter media for groundwater treatment and having the word “greensand” in their name, were analyzed. Their significant differences in compositional, structural and physical attributes were proven in work of Outram et al. [155]. Analysis of the mineral composition showed that two out of five have a quartz carrier, while none of the “greensands” contained glauconite. The carrier in other three materials was the mixture of manganese dioxide and montmorillonite as well as quartz and hematite. The coatings commonly called manganese dioxide was also investigated, and in their phase composition of pyrolusite, ramsdellite, romanechite and cryptomelane was detected [155].”

C23. L545-547 This sentence is very weird and needs to be better connected in this section.

A23. Rewritten

L 503-505: “The coatings commonly called manganese dioxide was also investigated, and in their phase composition of pyrolusite, ramsdellite, romanechite and cryptomelane was detected [155].”

C24. L579 what is the “manganese effect”?

A24. It is isunderstanding in translation, reworded to:

L 537-539: “Granops [156] describes the modernization of groundwater treatment plants, in which the use of Manganese Greensand significantly improved the manganese removal efficiency from both soft and hard water.”

C25. L581-582 I believed that larger hydraulic resistances would lengthen the filtration time?

A25. Of course, but during the filter run the flow rate is maintained on the level 4-12 m/h. Below the range the exploitation is rather uneconomical even though the contact time is longer. We have clarified the text:

L 539-542: “At high iron content (~5 mg/L), single-stage filtration on this type of bed is insufficient, as the fine granulation of the material hinders its operation under conditions of the formation the big amount of iron hydroxide clogging the bed. This leads to increase hydraulic resistances what requires more frequent backwashing and implies shortening of the filtration cycle.”

C26. L605-626 I don’t understand why is this section here. You compare the efficiency of commercial and non-commercial filters. It would be presented in a separated section.

A26. Klinopur-Mn and Klinomangan are the commercial products probably on the local market. We cannot find the commercial data sheets. For this reason, we have not provided in the table 4 information about these materials. We have changed the first introducing sentence in this paragraph also have reedited another:

L 566: “Other less common commercial products are coated zeolites Klinopur-Mn and Kilnomangan.”

L 571-575: “The mineral carrier of Klinomangan came from the Rátka deposit in Hungary and contained less clinoptilolite (55%) and more other ingredients such as cristobalite (15%), feldspar (10%) and montmorillonite (10%) [160]. After chemical coating Klinopur-Mn and Klinomangan contained 6.92 and 15.16 % of MnO2 respectively [160].”

C27. L628 and after, in this section you mention for the first time biological processes, yet it is not specific to naturally coated filter media and such phenomenon can occur with the other materials. I suggest to mention such phenomenon in the other sections.

A27. Please note, the first time we mentioned the biological treatment in Introduction chapter:

L 72-74: “Another mechanism for removing iron, manganese and mainly transforming ammonium ions is the biological oxidation by bacteria growing on the surface of the filter media [31].”

In case of your suggestion we have some doubts. Chemically coated beds are exploited using KMnO4 or NaClO (periodically or continuously). This practically prevents the development of microflora. Of course on the beds contained manganese ores the bacteria can grow, if the materials in filters are used without chlorination. Still, the manganese ores don’t need the start-up time for effective water treatment. The articles about biological treatment of groundwater on pyrolusite beds are not popular. We find the patented Pyrolusite Process® based on microorganism but consists of a limestone and applied for acidic water from mine drainage.

We also add the explanations in the article:

L 775-779: “In contrast to naturally coated filter media the manganese ores media don’t need the start-up time for effective water treatment. Even if biofilm develops on grains in favorable environment, biological oxidation is not the core of operation of these materials. In comparison to chemically coated filter media, the manganese ore can treat groundwater by catalytic oxidation without regeneration.”

C28. L632-633 “which participate in simultaneous transformation of iron and manganese [138,143,144] as well as iron, manganese and ammonium ions” please reword as it does not make any sense

A28. Our intention was to separate information about iron and manganese removal from iron, manganese and ammonium ions because it is important from the technological point of view. Two or three zones are formed in the filter bed profile (described next). We propose the following form:

L 593-595: “In this method spontaneous formation of coating occurs on the filter medium grains. Many authors report the use of naturally coated filter media for simultaneous removal of iron and manganese [164,169,170] as well as iron, ammonium ion and manganese [162,171–174].”

C29. L644-647 This conclusion/assumption is interesting but is not enoughly discussed in relation with the literature. Here you write X concludes … Y writes the opposite … Z writes another thing so I conclude… please provides understandable information to support your interesting conclusion.

A29. Of course, you are right. The text was unfortunate and inadequate. We changed it as follows:

L 607-614: “Also it has been proven that during the start-up period the oxidation is consistently of biological origin and over a prolonged filter run time of mature bed the removal based on physicochemical interactions [176] especially on adsorption and heterogeneous autocatalysis [167]. Biological treatment is possible even on ripened filter media and under slightly acidic conditions [177]. Quite dissimilar conclusions were drawn from the experiment of groundwater treatment on iron-manganese oxides coated filter media, the important role of ammonium ions and manganese transferring plays the abiotic catalytic oxidation [178].”

L 618-620: “These threads link to the conclusion, that it is very difficult to generalize which processes (biotic or abiotic) predominate in naturally coated filters and may result from individual specific operating conditions.”

C30. L648-649 “popular” “frequently used” in which context? Scientific literature? Field operation? Please be more specific

A30. reworded to:

L 621-622: “Quartz sand or gravel are the most popular materials used in field operations based on natural coating [29,165,166,176,178,182–184].”

C31. L650 I’m not sure that this section is the most appropriate to introduce a new material “melaphyre” please only cite those previously introduced.

A31. Of course, information about melaphyre was rejected.

C32. L652 “iron and manganese oxyhydroxides” in this section it may be highlighted that the natural coating is iron+manganese rather than only manganese. It is therefore hard to compare with lab-MnOx coated materials.

A32. Thank you for the idea, we summarized the section in this way:

L 737-742: “It should be emphasized that the heterogeneity of the chemical and phase composition as well as the morphological diversity of the naturally formed coatings make it very difficult to compare with chemically formed, obtained in laboratory or factory conditions. The dual nature of biotic and abiotic processes occurring on naturally coated filter media is an additional aspect that distinguishes them from chemically coated ones. Due to regeneration with oxidants, the commercial coated media purify water mainly based on abiotic way.”

C33. L655 “probably on properties” please reword this sentence

A33. reworded to:

L 626-628: “The duration of this period varies and depends on many factors, such as chemistry of groundwater, technological conditions of water filtration, filter design and properties of the filter medium, implies a rate of ripening [31,145]”

C34. L658-659 this sentence is too vague even it introduces an interesting assumption, please be clearer.

A34. We developed an argument:

L 629-634: “Duration of start-up period in many cases is brief (about 20 days) but can be also prolonged (many months), especially when chlorinated water is incorrectly used for backwashing, as in the last two examples. Systematic introduction of a chemical oxidant into the filter together with the backwashing water can be a factor that inhibits the growth of microorganisms. By some means, this observation can confirm the participation of microorganisms in the activation stage.”

C35. L674 “50-90%” really 90%?

A35. In the source is The sand grains were covered with a layer consisting mostly of manganese oxides

(characteristically black in colour) and iron oxides. The total thickness of the layer covering

the sand grains (0.8 to 1.6 mm diameter) was about 0.2 to 0.5 mm and corresponded to

approximately 50–90% of the total volume of the sand grains.”

We calculated it and the boundary values are 30-77%. I wonder what I can do in this case? I stay at the value given by authors. If you disagree, please correct us.

C36. L682-683 Chemically coated filter media does not present the potential for natural coating when used in field operations? Can you assume that chemically coated filter media used for groundwater treatment could be naturally coated by MnOx and FeOx? Please explain?

A36. Yes, you have a right. We added the explanation in text:

L 663-666: “It is likely that coatings on chemically coated media will also grow during the lifetime of the filter, although the authors cannot cite the confirmation. From practice it is known that excessive growth of the coating on grains is observed with improper filter operation like too low backwashing intensity.”

C37. L708 “iron and manganese removal zones” you use this expression several times throughout the manuscript, however, you never clearly explain this notion. As the readers will not be only specialists, a section explaining this may be useful (see comment above on other process)

A37. Of course, good idea. Brief explanation at the beginning of the section:

L 595-598: “In effect active oxidation zones are created in the filter bed profile. Referring to side of the filter supplied with aerated raw water, in the zones there are catalytic and bio-catalytic transformations of iron and manganese or iron, ammonia and manganese, respectively.”

C38. Table 5 Could be present Fe and Mn % in coatings? It would be more interesting than Na content?

A38. We added information in the table and made changes to the content of the paragraph:

L 687-695: “The elemental composition of formed coatings identified at various groundwater treatment plants is presented in Table 6. Iron and manganese, as well as oxygen from their oxyhydroxides, are the major elements. The surface of coated anthracite also contained carbon coming from biofilm or carrier [167]. The analytical method used has noticeable impact. The authors determined by extraction that 3-years old filter media contained 3.4 ±0.5 mg Fe/g and 5.0 ±0.3 mg Mn/g, while further ripening led to increase to 12.9 ±0.6 mg Fe/g and 19.9 ±3.3 mg Mn/g on 15-years old filter media [167]. Of the side elements, calcium and silica are the most common in coating, sometimes magnesium, phosphorus and aluminum, while sodium, potassium and sulfur are definitely less present (Table 6).”

At the end of the paragraph, we moved the sentence:

L 712-715: “A entirely dissimilar property is the homogeneity of sand surface after 20-day ripening obtained by EDS mapping. As the authors emphasize, it does not coincide with literature reports [182] and it must have been due to the specific conditions of formation..”

C39. Section 4, same comments on the structure and subsection (commercial/ores, experimental/operating) than for section 3

A39. Sections have changed - presented in answer 18

C40. Your conclusions are interesting but may be improved. I have the feeling that you are stick on very technical questions, and you did not conclude on the best materials for the groundwater treatment (even if I agree that such conclusion can not be such definitive).

A40. We added conclusion about removal of water constituents:

L 886-897: “As emphasized in opinions, groundwater treatment based on aeration and filtration on naturally coating media and on manganese ores is recommended due to the exclusion of chemical additives and prevents the formation of oxidation by-products. All of these MnOx filter media are able to remove manganese and iron from water based on various mechanisms. Manganese ore filter media are also capable of removing hydrogen sulphide and commercial chemically coated media additionally remove hydrogen sulphide, arsenic and radium. Purification of water from manganese, iron, ammonium ion, arsenic, phosphorus and antimony is possible using naturally coated filter media. However, the review shows that MnOx coated media obtained in laboratory conditions are highly effective in removing heavy metals and metalloids from water. Considering the above, the question may be asked to what extent technologies based on MnOx filter media can provide effective removal of heavy metals in operational conditions? This may be particularly important for groundwater treatment in degraded areas.”

  1. We also found small mistakes in table 7 and corrected it.

Reviewer 3 Report

This is interesting review about the MnOX and its derivatives used for groundwater treatment. 

Groundwater treatment is a vital topic for safe drinking water supply in the worldwide. The author put some efforts in the Poland areas, in fact more wide applications about MnOX or AlOx, are in the Asian areas, especially in the China and India. 

As thus I would like to suggest the author to add more literature and related studies about the research about Asian wide groundwater treatment. 

Secondly, besides the literature reviews, some practical performance evaluation, side effects and ions migration, and even its potential reactions with bacteria (from anaerobic environment to aerobic environment), how it will happen?

Thirdly, accompanying the stringent standards for drinking water supply, groundwater with lots of hidden pollutions injection is still the top priority as drinking water source? Can the author give some comments? or any potential treatment technologies advised for groundwater treatment.

Author Response

Dear Reviewer,

Thank you very much for your engagement in evaluation the submitted manuscript. We appreciate your different point of view at groundwater quality. In fact we think like Europe citizens. We have made changes and we hope that actual form of article is better. In second version of the manuscript the corrections to your comments are marked by green. The following answers are also included as an attachment.

Best regards,

Authors

C-Reviver’s comment

A-author’s answer

L-line number in corrected text

C1. This is interesting review about the MnOX and its derivatives used for groundwater treatment. Groundwater treatment is a vital topic for safe drinking water supply in the worldwide. The author put some efforts in the Poland areas, in fact more wide applications about MnOX or AlOx, are in the Asian areas, especially in the China and India. As thus I would like to suggest the author to add more literature and related studies about the research about Asian wide groundwater treatment.

A1. We were trying to broaden our view, so the second paragraph in introduction is rebuilt:

L 54-61: “According to the World Health Organization guidelines for drinking water quality [14], its components can not constitute a threat to human health. Groundwater can be a valuable resource of drinking water due to less pollution compared to surface water, though depending on the geological structure, hydrogeological and climatic conditions, in some regions of the world with intensive agricultural, urban and industrial land use [15–17] it is already contaminated with heavy metals and metalloids, nitrate, organics or halides [18–27]. Iron, manganese, ammonium ion and hydrogen sulphide are common components of groundwater and can be removed by classic technologies based on manganese oxides coated and consisted of filter media.”

We also added the references from Asia in the topic:

Coyte, R.M.; Singh, A.; Furst, K.E.; Mitch, W.A.; Vengosh, A. Co-occurrence of geogenic and anthropogenic contaminants in groundwater from Rajasthan, India. Sci. Total Environ. 2019, 688, 1216–1227, doi:10.1016/j.scitotenv.2019.06.334.

Vatandoost, M.; Naghipour, D.; Omidi, S.; Ashrafi, S.D. Survey and mapping of heavy metals in groundwater resources around the region of the Anzali International Wetland; a dataset. Data Brief 2018, 18, 463–469, doi:10.1016/j.dib.2018.03.058

Li, F.; Qiu, Z.; Zhang, J.; Liu, W.; Liu, C.; Zeng, G. Investigation, Pollution Mapping and Simulative Leakage Health Risk Assessment for Heavy Metals and Metalloids in Groundwater from a Typical Brownfield, Middle China. Int. J. Environ. Res. Public. Health 2017, 14, 768, doi:10.3390/ijerph14070768.

Liang, C.-P.; Jang, C.-S.; Liang, C.-W.; Chen, J.-S. Groundwater Vulnerability Assessment of the Pingtung Plain in Southern Taiwan. Int. J. Environ. Res. Public. Health 2016, 13, 1167, doi:10.3390/ijerph13111167.

Li, J.; He, Z.; Du, J.; Zhao, L.; Chen, L.; Zhu, X.; Lin, P.; Fang, S.; Zhao, M.; Tian, Q. Regional Variability of Agriculturally-Derived Nitrate-Nitrogen in Shallow Groundwater in China, 2004–2014. Sustainability 2018, 10, 1393, doi:10.3390/su10051393.

Guo, Q.; Zhou, Z.; Huang, G.; Dou, Z. Variations of Groundwater Quality in the Multi-Layered Aquifer System near the Luanhe River, China. Sustainability 2019, 11, 994, doi:10.3390/su11040994.

Ahmad, A.Y.; Al-Ghouti, M.A. Approaches to achieve sustainable use and management of groundwater resources in Qatar: A review. Groundw. Sustain. Dev. 2020, 11, 100367, doi:10.1016/j.gsd.2020.100367.

Jia, H.; Qian, H.; Qu, W.; Zheng, L.; Feng, W.; Ren, W. Fluoride Occurrence and Human Health Risk in Drinking Water Wells from Southern Edge of Chinese Loess Plateau. Int. J. Environ. Res. Public. Health 2019, 16, 1683, doi:10.3390/ijerph16101683.

as well as from other regions of the world:

Verlicchi, P.; Grillini, V. Surface Water and Groundwater Quality in South Africa and Mozambique—Analysis of the Most Critical Pollutants for Drinking Purposes and Challenges in Water Treatment Selection. Water 2020, 12, 305, doi:10.3390/w12010305.

Lapworth, D.J.; Baran, N.; Stuart, M.E.; Ward, R.S. Emerging organic contaminants in groundwater: A review of sources, fate and occurrence. Environ. Pollut. 2012, 163, 287–303, doi:10.1016/j.envpol.2011.12.034.

Koda, E.; Sieczka, A.; Miszkowska, A.; Osiński, P. Groundwater Contamination by Organic Compounds: A Case Study of Łubna Landfill Site in Warsaw, Poland. In Environmental Geotechnology; Agnihotri, A.K., Reddy, K.R., Bansal, A., Eds.; Lecture Notes in Civil Engineering; Springer Singapore: Singapore, 2019; Vol. 31, pp. 307–317 ISBN 9789811370090.

We have also added the following references to Asian authors writing about water treatment:

Zhou, D.; Kim, D.-G.; Ko, S.-O. Heavy metal adsorption with biogenic manganese oxides generated by Pseudomonas putida strain MnB1. J. Ind. Eng. Chem. 2015, 24, 132–139, doi:10.1016/j.jiec.2014.09.020.

Han, M.; Zhao, Z.; Gao, W.; Cui, F. Study on the factors affecting simultaneous removal of ammonia and manganese by pilot-scale biological aerated filter (BAF) for drinking water pre-treatment. Bioresour. Technol. 2013, 145, 17–24, doi:10.1016/j.biortech.2013.02.101.

Zhao, Y.; Tian, G.; Duan, X.; Liang, X.; Meng, J.; Liang, J. Environmental Applications of Diatomite Minerals in Removing Heavy Metals from Water. Ind. Eng. Chem. Res. 2019, 58, 11638–11652, doi:10.1021/acs.iecr.9b01941.

Jiang, N.; Shang, R.; Heijman, S.G.J.; Rietveld, L.C. High-silica zeolites for adsorption of organic micro-pollutants in water treatment: A review. Water Res. 2018, 144, 145–161, doi:10.1016/j.watres.2018.07.017.

C2: Secondly, besides the literature reviews, some practical performance evaluation, side effects and ions migration, and even its potential reactions with bacteria (from anaerobic environment to aerobic environment), how it will happen?

A2. Redox potential Eh and pH are the main factors that determine whether precipitation will take place biotically or abiotically. Biological iron removal occurs at a pH above 7 and a redox potential value of 10 mV. In addition, biological ferrous iron is characterized by the fact that it can take place even at extremely low oxygen content from 0.005 to 0.025 mg O2 / dm3. Biological oxidation of manganese occurs at relatively high value of redox potential - about +600 mV at pH 6.5 7.0. Manganese oxidizing bacteria belong to microaerophiles, and therefore their optimal development occurs under conditions of limited oxygen dissolved in water, as well as they are classified as organisms developing in a transition environment. In text we added general information about factors affecting removal and nature of bacteria:

L 601-604: “Unlike aerobic nitrifying bacteria, manganese and iron oxidizing bacteria belong to microaerophiles, and therefore their optimal development occurs under conditions of limited oxygen dissolved in water, as well as they are classified as organisms developing in a transition environment.”

L 616-618: “Groundwater composition, its Eh, pH, temperature and oxygen concentration, but also flow velocity, media type and backwashing method are the main factors for treatment of aerated water by filtration [31] and can determine the nature of the process.”

C3. Thirdly, accompanying the stringent standards for drinking water supply, groundwater with lots of hidden pollutions injection is still the top priority as drinking water source? Can the author give some comments? or any potential treatment technologies advised for groundwater treatment.

A3. Generally the second paragraph of introduction chapter was reorganized. We determined that iron, manganese, ammonium ion and hydrogen sulphide are the target components for technologies using described materials:

L 59-61: “Iron, manganese, ammonium ion and hydrogen sulphide are common components of groundwater and can be removed by classic technologies based on manganese oxides coated and consisted of filter media.”

Of course we know several methods and complex technologies for removal anthropogenic pollutants from groundwater. We remind these impurities according to your first comment. However, we believe that describing solutions for removing other contaminants will cause a significant departure from the topic. We hope that the additional paragraph attached to the section “Conclusions” can partially answer this point:

L 886-897: “As emphasized in opinions, groundwater treatment based on aeration and filtration on naturally coating media and on manganese ores is recommended due to the exclusion of chemical additives and prevents the formation of oxidation by-products. All of these MnOx filter media are able to remove manganese and iron from water based on various mechanisms. Manganese ore filter media are also capable of removing hydrogen sulphide and commercial chemically coated media additionally remove hydrogen sulphide, arsenic and radium. Purification of water from manganese, iron, ammonium ion, arsenic, phosphorus and antimony is possible using naturally coated filter media. However, the review shows that MnOx coated media obtained in laboratory conditions are highly effective in removing heavy metals and metalloids from water. Considering the above, the question may be asked to what extent technologies based on MnOx filter media can provide effective removal of heavy metals in operational conditions? This may be particularly important for groundwater treatment in degraded areas.”

Round 2

Reviewer 2 Report

The authors have performed very significant efforts for the revision of their manuscript according to all the comments of the reviewers. I congratulate the authors for that and strongly recommend their manuscript for publication as revised. Please just look out for the repetition of the numbering of references in the last section.

Reviewer 3 Report

The revised manuscript is ready for publication.